# Exploring the conditions conducive to convection within the Greenland Ice Sheet

Robert Law[1, 2, 3, 4], Andreas Born[1, 2], Philipp Voigt[1, 2], Joseph A. MacGregor[5], and Claire Marie Guimond[6]

[1]Department of Earth Sciences, University of Bergen, Norway
[2]Bjerknes Centre for Climate Research, Bergen, Norway
[3]Laboratory of Hydraulics, Hydrology and Glaciology (VAW), ETH Zurich, Zurich, Switzerland
[4]Swiss Federal Institute for Forest, Snow and Landscape Research (WSL), bâtiment ALPOLE, Sion, Switzerland
[5]Cryospheric Sciences Laboratory, NASA Goddard Space Flight Center, Greenbelt, USA
[6]Atmospheric, Oceanic, and Planetary Physics, University of Oxford, UK

**Correspondence:** Robert Law (roblaw@ethz.ch)

**Abstract.** Enigmatic large plume-like features disrupt the radiostratigraphy within the Greenland Ice Sheet. Here we use the ASPECT geodynamics modelling package to test the hypothesis that convection is a viable mechanism for the formation of the large (>1/3 ice thickness) englacial plume-like features observed in north Greenland, provided that there is a modest initial temperature perturbation. Both greater horizontal shear and snow accumulation impede formation of convection plumes, while low shear and softer ice encourages them. These results potentially explain the dearth of larger basal plumes in the younger and higher-accumulation southern ice sheet. We leverage this apparent convection mechanism to place bounds on ice rheology, which suggests that – for parts of north Greenland – effective ice viscosity may span $\sim 2 \times 10^{12}$-$3 \times 10^{14}$ Pa s, or around an order of magnitude lower than commonly assumed. Softer ice there implies reduced basal slip compared to standard models. Isolating if this effective viscosity range is impacted by additional processes (like basal freeze on and travelling slippery patches) and implementing a softer basal ice rheology in numerical models may help reduce uncertainty in projections of future ice-sheet mass balance.

## 1 Introduction

The Greenland Ice Sheet (GrIS) is a major cryospheric contributor to global sea level rise (Otosaka et al., 2023), with numerical models predicting accelerating, although uncertain, GrIS mass loss throughout the 21[st] century and beyond (Aschwanden and Brinkerhoff, 2022). However, numerous aspects of the GrIS's thermodynamics and hence motion remain enigmatic, including the widespread presence of large (greater than 1/3 of total ice thickness) englacial plumes found by tracing reflections of equal age in radargrams (i.e., isochrones; Figs. 1A, 2, A1; Bell et al., 2014; CReSIS, 2013). Mapping (Leysinger Vieli et al., 2018) and visual inspection of automatically-tracked disrupted radiostratigraphy (Panton and Karlsson, 2015) shows that these large plume-like features (hereafter plumes) are mostly found in the northern part of the GrIS (Figs. 1A, A1) but a consensus formation mechanism has not yet been identified. Although the plumes themselves are unlikely to be critical in interpretation of ongoing mass loss processes, clarifying their formation mechanism may reveal important information about the rheology,

basal thermal state, and stability of the locations where they are or are not found – ultimately improving representation of ice flow in model projections.

These plumes have previously been hypothesized to result from basal freeze on (Leysinger Vieli et al., 2018), or travelling basal slippery spots (Wolovick et al., 2014), which both require that the bed be at least locally or temporarily thawed. Separately, Bons et al. (2016) and Zhang et al. (2024b) show that convergent flow, rheological anisotropy, and a rough bed are sufficient to form small-scale (<100 m) folds, but that, when basal slip and freeze on processes are excluded, large-scale folds require density gradients induced by thermal expansion and significantly lower viscosity. Another way to describe such temperature- and buoyancy-driven fold formation – and which lies along the same process continuum – is convection. In thermal convection, layers of ice heated geothermally from below (or cooled from above) thermally expand at the bottom (or thermally contract at the top), creating an unstable density gradient and forcing a vertical flow of material.

Here, we use 'local convection' to refer to a temperature- and density-controlled process generating self-sustaining upwards motion and disrupted plume-like structures emanating from the bed, superimposed upon primary thermodynamic processes driving interior ice towards the ice sheet's margins, that are relatively isolated spatially. We explore whether ice convection – a process with a contentious history in theoretical glaciology – can explain observations of these large plumes, also known as disrupted basal units. Hughes (1976, 2012) previously proposed convection within ice sheets, but only for full-thickness con- vection (rather than stagnant-lid) and attracting strong objection (Fowler, 2013). Both Hughes and Fowler approach convection analytically only, by estimating a Rayleigh number $Ra$, the dimensionless ratio of heat transfer via upwards mass transport (i.e. convection) vs. thermal conduction (Appendix A1, Rayleigh, 1916). In these analytical models, convection initiates when a critical Rayleigh number is exceeded ($\sim 650 - 1700$ in Knopoff, 1964 and Hughes, 1976), reached already if a 2,500 m ice ice column has a uniform effective viscosity below $4 \times 10^{14}$ Pa s (see below and Appendix A1 for further information on effective viscosity). While Hughes and Fowler found $Ra$ values close enough to the critical value to warrant consideration of convection, applying a purely analytic approach to the GrIS is not ideal. The formulation of thermal diffusion in $Ra$ does not capture dynamical effects important in ice sheet flow, such as horizontal shearing or downwards motion from snowfall; and the critical $Ra$ value is itself tied to the particular boundary conditions and the initial perturbation geometry (e.g., Solomatov, 1995) making an analytical approach challenging for a dynamically and materially complicated system perched close to the onset of convection behaviour. We therefore consider a direct numerical modelling approach more appropriate for investigating the question of convection in terrestrial ice sheets.

## 2 Methods

We use the geodynamics software package ASPECT 2.5.0 (Kronbichler et al., 2012; Heister et al., 2017; Bangerth et al., 2023) to test our convection hypothesis. The setup is adjusted to simulate a 25 km along-flow two-dimensional slice through an ice sheet (Fig. 3), or a 22 km along-flow by 18 km across-flow three-dimensional cuboid (Fig. A4). ASPECT is used in place of a conventional ice sheet model due to its extensive benchmarking in convection problems and built-in functionality to model buoyancy forces, which are lacking in modern ice-sheet models. Similar geodynamics models have been previously used to

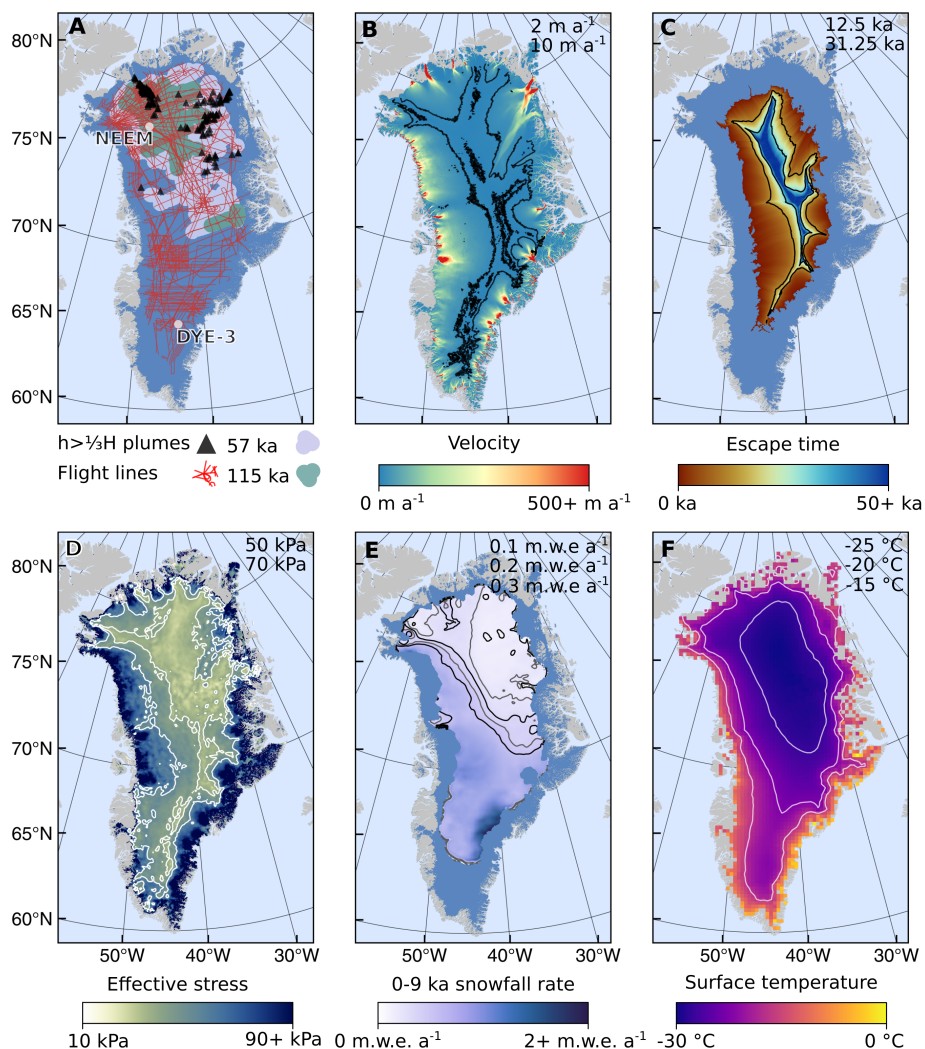

**Figure 1.** Relevant GrIS properties when considering convection. Contour values are given in the top right of each panel. **A** Location of large plumes from Leysinger Vieli et al. (2018), NEEM and DYE-3 boreholes, flight lines, and detection of 57 and 115 ka age ice layers from Macgregor et al. (2015). **B** Surface velocity using NASA MEaSUREs ITS_LIVE velocity data. **C** Escape time required to reach the 2,000 m ice-thickness contour using the same source data as **B** and a shape factor of 0.8 to better approximate column-averaged velocity (e.g. Whillans, 1977). **D** Effective stress at 5/14 depth obtained from ISSM run. Fig. A2 shows a 3D view of this effective stress. **E** Averaged accumulation rate from MacGregor et al. (2016) for 0-9 ka. The two grey lines represent contours of 0.15 and 0.25 m.w.e a$^{-1}$. **F** Mean annual temperature from RACMO averaged over 1959-2019 (Noël et al., 2018). Background data from QGreenland (Moon et al., 2022)

.

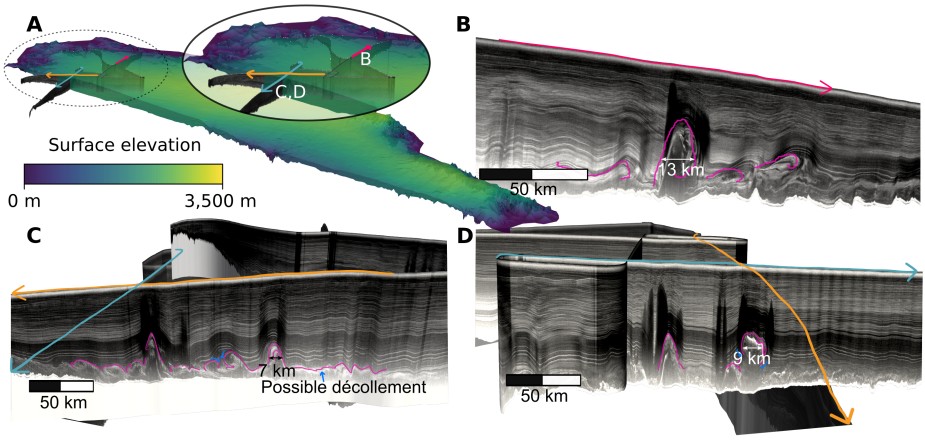

**Figure 2.** Example plume features in north Greenland. **A** Oblique view of the GrIS from the southwest with surface elevation from BedMachine (Morlighem et al., 2017) with a vertical exaggeration of 25 (and also in panels B-D). **B**, **C**, and **D** detail of plumes corresponding to coloured arrows in **A** with data from CReSIS (2013).

study convection in the shells of icy moons (Lebec et al., 2023). To facilitate a broad parameter sweep at low computational expense, and to isolate the influence of the parameters in question, we simplify the domain to have a uniform ice thickness (2.5 km as a reference value).

For all simulations except those that explicitly consider snowfall surface mass balance is set to zero, i.e., no snowfall or surface melting ($v_{z,s} = 0$ at the surface boundary condition where $\boldsymbol{v} = (v_x, v_y, v_z)$ is the velocity field, the subscript $x$ represents the along-flow distance, the subscript $z$ is depth, and the subscript $y$ is across-flow distance in 3-D simulations). When applied, surface shearing velocity $v_{x,s}$ is uniform across the domain's top surface with the rigid $v_{x,b} = 0$ condition maintained at the base. This Dirichlet velocity condition on a fixed surface differs from a 'standard' ice sheet model, where surface velocity is an emergent result of ice geometry and flow parameters, but is suitable for our purposes as we treat surface velocity as an independent variable in simulations. Regardless, the net effect on background (i.e., not convection controlled) stress and strain fields is similar. Keeping $v_{x,b} = 0$ is likely a firmer control on basal velocity than the possible décollement observed in radargrams (Fig. 2C). Therefore, while increasing $v_{x,s}$ in the model can simulate plume behaviour as *actual* surface velocity increases (Fig. 1C), the comparison is not one-to-one . In reality, surface displacement could also be accommodated by basal slip or a thin shear layer beneath the plumes (perhaps visible in Fig. 2) meaning our modelled plume behaviour represents the lower limit of stratigraphic disruption under a given surface velocity. Similarly, the placement of the initial perturbation in snowfall runs will influence the balance between horizontal and vertical velocity components (Fig. A5). Further, snowfall, surface velocity, and ice thickness all exhibit moderate variation over the millennial timescales important for convection (MacGregor et al., 2016). Resolution is determined by the ASPECT requirement to set grid spacings as a given number of even divisions. In the case of a 2,500 m thickness and 6 divisions this gives horizontal and vertical resolutions of 390 and 39 m, respectively.

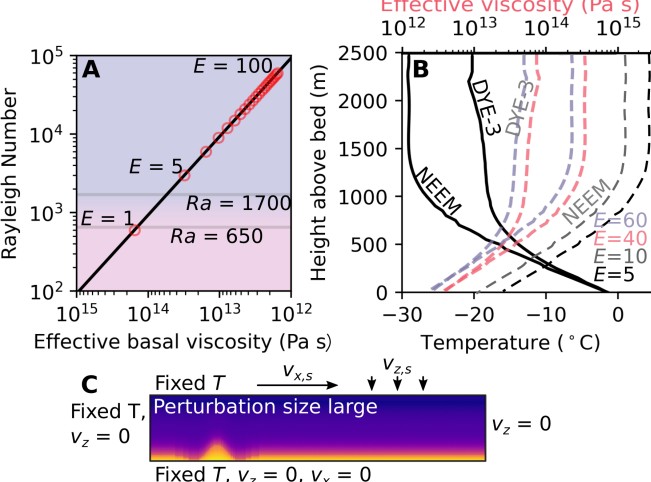

**Figure 3.** Parameter and model information. **A** Rayleigh number $R_a$ calculated assuming the basal effective viscosity is constant through an ice column of 2,500 m thickness and how enhancement factors, $E$, relate to a given effective basal viscosity with 'standard' parameters (Table A1 from Cuffey et al., 2000) when $\tau_e = 5 \times 10^4$ Pa. The lower and upper gray lines show $R_a$ values of 650 and 1700, respectively. **B** Temperature profiles from DYE-3 and NEEM (Dahl-Jensen et al., 2013; Rasmussen et al., 2013) and equivalent effective viscosity profiles given enhancement factors of 40 (red) and 60 (blue). **C** Initial 2D model domain showing boundary conditions with a vertical exaggeration of 2, length of 25 km and height of 2,500 m and a large temperature perturbation for the NEEM profile. The plasma colormap corresponds to the temperature colorbar in Figs. 1, 6 The medium temperature perturbation and 3D setup are shown in Fig. A4.

In most ice-sheet models, stress is related to strain rate with the Nye–Glen isotropic flow law (Nye, 1952; Glen, 1955)

$$\dot{\boldsymbol{\epsilon}} = A \tau_e^{n-1} \boldsymbol{\tau}, \tag{1}$$

where $\dot{\boldsymbol{\epsilon}}$ is the strain-rate tensor, $\boldsymbol{\tau}$ is the deviatoric stress tensor, $\tau_e^2 = \frac{1}{2}\mathrm{tr}(\boldsymbol{\tau}^2)$ defines the effective stress $\tau_e$ (Pa), $n$ is the flow exponent generally assumed as 3 or 4 (Cuffey and Paterson, 2010; Bons et al., 2018), and $A = EA_0\exp\left(-\frac{Q}{R}\left(\frac{1}{T} - \frac{1}{T_0}\right)\right)$ (Pa$^{-n}$ a$^{-1}$) is the creep parameter, where $E$ is the enhancement factor, $A_0$ is the creep prefactor, $Q$ (J mol$^{-1}$) is the activation energy, $R$ is the ideal gas constant (J mol$^{-1}$ K$^{-1}$), $T_0 = 263.2$ K, and $T$ (K) is ice temperature. $E$ is defined as $\dot{\epsilon}_m/\dot{\epsilon}_o$ where $\dot{\epsilon}_m$

is the measured strain rate and $\dot{\epsilon}_o$ is the strain rate predicted by Eq. 1. The value and influence of $E$ therefore varies depending on the choice of $A_0$ and $n$; we use the default values in Cuffey and Paterson (2010) as a widely used reference (Table A1), which makes comparison with existing ice-sheet models more straightforward. $E$ varies based on deformation type but is often assumed to be $\sim$4-6 for the GrIS when using $n = 3$, though it has been inferred to be up to 12 in Antarctic shear margins (Echelmeyer et al., 1994) and even 120 in mountain glaciers (Echelmeyer and Zhongxiang, 1987). In most ice-sheet models,

separate $Q$ values are used for high ($T > T_0 \approx 263.1$ K) and low temperatures, but for simplicity in ASPECT here we simplify this to one mid-range value, which has limited effect (Fig. A3). Moreover, the pressure dependence of $A$ is neglected.

We also simplify to a Newtonian rheology for ASPECT by setting $\tau_e^{n-1}$ constant. We use $n = 3$ with $\tau_e = 50$ kPa as a reasonable starting value (Figs. 1D, A2) based on an Ice-sheet and Sea-level System Model simulation (Larour et al., 2012) with

further information, and justification for a constant $\tau_e$ value provided in Appendix A2. A Newtonian rheology is appropriate here as strain rates due to convection are small compared to those from background ice flow (Fig. A5), and convection can be considered a secondary phenomenon in this sense, though the implications of the rheological setup and choice of $\tau_e$ are covered further in the Discussion and Appendix A2. Prescribing $\tau_e$ is also necessary as a full ice-sheet stress state can not be accurately replicated in a simplified along-flow slice. Effective viscosity $\eta$ can then be calculated as

$$\eta = \frac{1}{2}\left[A\tau_e^{n-1}\right]^{-1}. \tag{2}$$

Rearranging Eq. 2 yields the functional dependence of $E$:

$$E = c\eta^{-1}\tau_e^{1-n}, \tag{3}$$

where $c = (2A_0)^{-1}\exp\left(\frac{Q}{R}\left[T^{-1} - T_0^{-1}\right]\right)$. The temperature-dependent viscosity is then controlled by varying $E$ (Fig. 3) as a tool to test rheological variation. We furthermore describe some results using $E$ but note that the *actual* effective viscosity remains a complex function of ongoing and historic stress and deformation states.

ASPECT solves the governing equations of convection,

$$\nabla \cdot \boldsymbol{v} = 0 \qquad \text{(conservation of mass)} \tag{4}$$

$$-\nabla \cdot [2\eta\dot{\boldsymbol{\epsilon}}] + \nabla p' = -\beta\bar{\rho}T'g \qquad \text{(conservation of momentum)} \tag{5}$$

$$\bar{\rho}C_p\left(\frac{\partial T}{\partial t} + \boldsymbol{v} \cdot \nabla T\right) - \nabla \cdot \kappa\nabla T = F_{\text{int}} \qquad \text{(conservation of energy)} \tag{6}$$

where $p$ (Pa) is pressure, $C_p$ (J kg$^{-1}$ K$^{-1}$) is heat capacity, $\kappa$ (W m$^{-1}$ K$^{-1}$) is thermal conductivity, $\beta$ (K$^{-1}$) is thermal expansion coefficient, $\bar{\rho}$ (kg m$^{-3}$) is the reference density, $g$ (m s$^{-2}$) is acceleration due to gravity, and $F_{\text{int}}$ (W m$^{-2}$) is the sum of all other heating terms. We set $F_{\text{int}} = 0$, thereby ignoring adiabatic heating and neglecting strain heating, to prevent simulations with greater $v_{x,s}$ and hence greater strain heating from evolving a different rheology along flow, though we note that such heating will soften ice and may further facilitate convection. These equations follow the Boussinesq approximation – that density variations are small enough to be neglected everywhere except for in the buoyancy term $\beta\bar{\rho}T'g$ – which is valid for very slow-flowing materials without abrupt density changes. This solution method simplifies the temperature field to $T = \bar{T} + T'$, where $\bar{T}$ is a constant reference temperature and $T'$ is the temperature perturbation; analogous perturbations are formed for the pressure and density fields.

Two baseline temperature profiles are used, representing the colder NEEM ice-core site in northern Greenland (Dahl-Jensen et al., 2013; Rasmussen et al., 2013) and the warmer DYE-3 in southern Greenland (Gundestrup and Hansen, 1984) respectively (Figs. 1A, 3). We apply a transformation, $T_2 = \frac{T_1 + T_a}{T_b}(T_b + T_a)$ where $T_1$ is the original temperature profile, $T_b$ is the basal temperature and $T_a$ is an adjustment term used to raise the basal temperature to $-2^oC$. The temperature profiles are stretched and compressed when adapted to the range of ice thicknesses. This is not a perfect representation of ice-sheet ice temperature, but allows more direct comparison between simulations of different thicknesses, and other temperature estimation methods are subject to their own uncertainties. As we keep the basal temperature uniform across the domain (Dirichlet condition, therefore

indirectly neglecting geothermal heat flux which nonetheless modulates through a realistic range over our simulations; see Results, Fig. A6) and also want to consider ice some distance from the ice core site, a slightly higher fixed basal value is appropriate as a midpoint between the ice-sheet interior and margins. Above ∼3,000 m, the basal temperature is then technically above the pressure-melting-point, even though a non-slip condition is imposed at the base at all times. Simulations where $H >$3,000 m comprise a small proportion of our overall ensemble and we do not change the basal temperature for this subset.

An initial temperature perturbation replicating a fold is created 5 km in from the inflow side (3.5 km in the case of snowfall simulations) using two Gaussian functions of opposing signs. We refer to two temperature perturbation sizes for 2D runs: large, used for most simulations (Fig. 3), and medium (Fig. A4). In the 3D runs a simpler approach is taken, with a cube of uniform 273 K ice measuring 3,000 m × 5,000 m × 750 m as the initial perturbation. As the initial temperature gradient is not linear (Fig. 3), using a larger initial perturbation allows convection to occur in a more realistic temperature field without a delay for

initial plume development, though results are relatively insensitive to the initial perturbation (Figs 4B, 4C, A6). ASPECT input files and scripts to recreate the temperature perturbations and perform other post-processing operations are provided in the Open Research Section. Values for set parameters are given in Table A1 and Figs. 3 and A4 illustrate boundary conditions.

    For each temperature profile we focus on the influence of four variables on the maximum upwards-directed vertical velocity, $\max(v_z)$: the enhancement factor ($E$), shear velocity ($v_{x,s}$), ice thickness ($H$), and snow accumulation rate ($v_{z,s}$) (labelled

in Table A2, Fig. 4). Additional 3D simulations are included for runs B and F, which consider the parameter space covering observed plumes (NEEM temperature profile). Defining a threshold for convection under a given parameter space is not straightforward, but we focus on $\max(v_z)$ over time as a reasonable indicator. Nonetheless, even if $\max(v_z)$ trends towards zero over time, the englacial stratigraphy will still be slightly disrupted during this transition period. The total buoyancy forces in the 3D simulation will also be greater than in 2D as we are able to model an isolated plume rather than a laterally extensive

140   fold. We divide behaviour into three zones, focusing on the 2D simulations that cover a broader parameter space. **Suppressed** convection is defined where $\max(v_z)$ at 20 kyr is below 0.01 m yr$^{-1}$ or where $\max(v_z)$ at 20 kyr has dropped by 0.03 m yr$^{-1}$ or more relative to its value at 4 kyr. **Amplifying** convection is defined where $\max(v_z)$ at 20 ka exceeds 0.4 m yr$^{-1}$ or where $\max(v_z)$ has increased by 0.1 m yr$^{-1}$ or more between 4–20 kyr ka. **Sustained** convection then occupies the space between these two zones. This approach allows us to isolate which parameter combinations may produce sufficient upwards flow to

account for the distribution of large englacial plumes (Fig. 1A).

## 3   Results

While our modelling is substantially more sophisticated than calculation of a single $Ra$ value, the general behaviour in our simulations can still be understood in terms of the $Ra$ number (Appendix A1), with greater values of $E$ and $H$ prompting convection. Given the additional complications of varying surface velocity, snowfall, initial perturbation, and viscosity profile,

we find that there is no single critical value of $E$ that describes this transition, but Figs. 4, 5 suggest that, for $\tau_e = 50$ kPa, $45 \leq E \leq 75$ encapsulates a range of behaviour for the NEEM temperature profile sufficient to form features similar to those observed in radiostratigraphy (Fig. 6). Considering shear over the domain of 1 m yr$^{-1}$ with no snowfall (Fig. 4B) in 3D,

$\max(v_z)$ begins to increase from 4–14 kyr between $40 < E < 50$, around the same point at which 2D convection is considered to be sustained under our definition. In Fig. 4B at $E = 75$, $\max(v_z)$ is consistently increasing over time and when $E = 60$ convection is still classified as sustained for snowfall rates exceeding 0.15 m yr$^{-1}$. Fig. 5 shows suppressed, amplifying, and sustained behaviour as a time series under different scenarios.

DYE-3 requires much lower $E$ values to transition between suppressed, sustained, and amplifying zones compared to NEEM when other parameters are equivalent (Fig. 5a; cf. Figs. 4A, B and Figs. 6A, C). The steep temperature gradient at the base of the DYE-3 profile is sustained over a shorter height, resulting in lower buoyancy forces overall, but this is compensated by lower viscosity in the upper portion of the domain (Fig. 3). Ice thickness is an important factor for both profiles, with sustained convection becoming infeasible below thicknesses of around 2,000 m (hence its use as a boundary in Fig. 1B), though note that ice thickness is also intertwined to an extent with its influence on the basal temperature gradient. Vertical transport rates begin to level off with increasing ice thickness for the DYE-3 profile, pointing towards a maximum rate of upwards motion. However, surface velocity and snowfall exert perhaps the most important overall constraints on $\max(v_z)$, with convection becoming infeasible as surface velocity in our simulations increases from 1 to 3 m yr$^{-1}$ or as snowfall increases beyond 0.1 to 0.3 m yr$^{-1}$, with the precise cut-off value depending on $E$ and the temperature profile (Figs. 4D-F, J-L).

The modelled plume geometry varies substantially across the parameter space. In the suppressed convection zone, a perturbation is still produced but is not sustained (Figs. 4D, 6D). Amplifying convection can prompt a self-sustaining plume chain with significant temperature variation (Figs. 4A, 6C). Sustained convection (Figs. 4B, F, L, 6A, B, E) produces plumes more similar to the folds observed in radargrams, though plume length is slightly shorter. 3D simulations produce roll-over in the down-flow direction, but characteristic symmetric overturning typical of convection in an otherwise static medium in the across-flow direction (Figs. 2D, 6E). Using a smaller perturbation in 2D simulations does not appreciably alter this pattern, but does slightly and consistently shift down the maximum $v_z$ over the simulation time frame (Fig. 5a). The heat flux required to sustain the fixed basal temperature ($\sim$40-70 mW m$^{-1}$, Fig. A6) is compatible with proposed rates of geothermal heat flux beneath the GrIS (Zhang et al., 2024a).

## 4  Discussion

Our results suggest five main thresholds for convection to occur within the Greenland Ice Sheet: (1) an initial temperature and therefore density perturbation; (2) ice thickness must be greater than around 2,200 m; (3) snowfall must be less than around 0.15 m yr$^{-1}$; (4) total horizontal shear through the column convection is occurring in must be less than around 1 m yr$^{-1}$; and (5) the effective viscosity profile range should fall within $\sim$2$\times$10$^{12}$-3$\times$10$^{14}$ (equivalent to an enhancement factor of $\sim$45-75 if $\tau_e = 50$ kPa) – roughly an order of magnitude lower than may typically be anticipated (Fig. 3). Condition (1) is easily satisfied by bedrock perturbations (e.g. Figs. 2B, C) or basal folding induced by processes such as convergence and rheological anisotropy (Zhang et al., 2024b); condition (2) is satisfied for a large region of the interior of the central ice sheet (Fig. 1B); and condition (3) is primarily satisfied in north Greenland (Fig. 1E). Condition (4) is satisfied by low surface velocities throughout large parts of northern central Greenland (Fig. 1C), with longer residence time being unique to the northern central ice divide

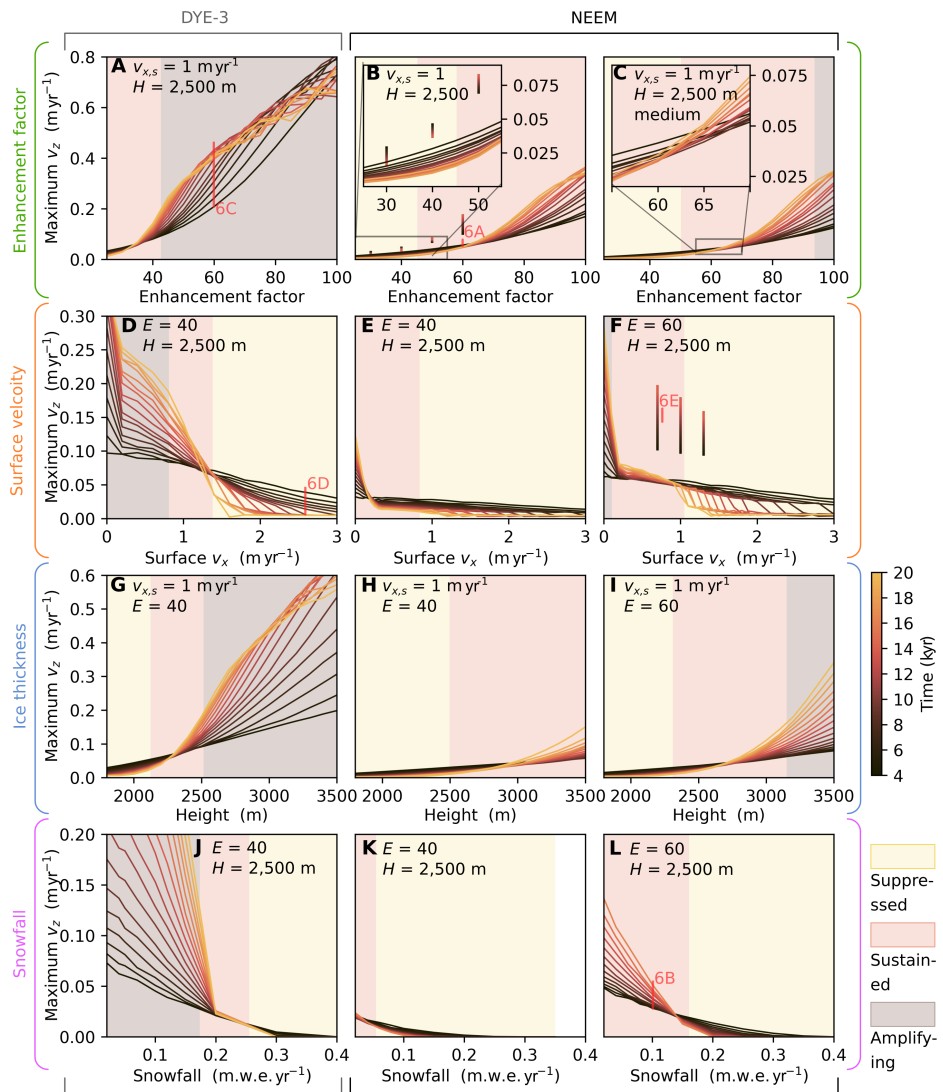

**Figure 4.** Convection model ensemble results. Relevant model parameters are given in the top left of each panel and along the left and top figure border. We display two columns for the NEEM temperature profile – beyond the first row these show a difference in $E$ between 40 and 60, while the first row shows the difference between the large (left) and medium (right) temperature perturbations. All runs use the large initial perturbation except for **C**. Red lines and letters in **A**, **B**, **D**, and **F** refer to the runs shown in the correspondingly labelled panels of Fig 6 and vertical lines in **B** and **F** correspond to 3D simulations. All simulations except for those with non-zero snowfall (**J**, **K**, **L**) have $v_{z,s} = 0$. An effective stress, $\tau_e$, or 50 kPa is used in calculating all effective viscosity profiles. See the text for our definitions of Suppressed, Sustained, and Amplifying convection, printed in bold for easy visibility. Note panels **K** and **L** only run to 16 kyr.

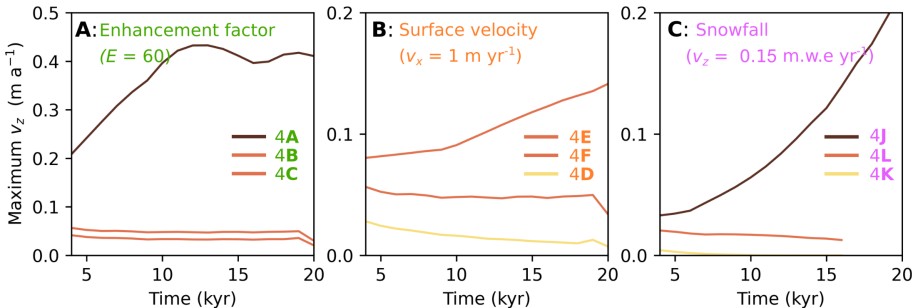

**Figure 5.** Time series of maximum upwards oriented velocity. **A** shows data from panels 4A, 4B, and 4C with an enhancement factor of 60. **B** shows data from panels 4D, 4E, and 4F with a surface velocity of 1 m yr⁻¹ (meaning the lines for 4B and 4F are the same). **C** shows data from panels 4J, 4K, and 4L with snowfall of 0.15 m.w.e. yr⁻¹. Labelling in each panel runs from topmost to bottommost line. 4L only runs to 16 kyr (and 4K goes to zero before this point).

(Fig. 1B); and , MacGregor et al., 2016). However, observed plumes are always located in regions where surface velocity is >1 m a⁻¹ and sometimes found where surface velocity is >10 m a⁻¹. In part, this may arise from surface velocity being mitigated by basal slip and shear beneath the plumes (see also Methods) but this issue is addressed further as we progress through the discussion.

We consider the possibility that effective viscosities for Greenland may be lower than generally assumed for north Greenland, satisfying condition (5), after first discussing other aspects of plume morphology and distribution. First, modelled plume widths (Fig. 6) are comparable but slightly narrower than observations. This may occur due to the continued disruption of plumes as the velocity field evolves after they have attained their maximum amplitude in a thicker, colder, slower, and larger palaeo-GrIS (Lecavalier et al., 2014). However, this places plumes in a delicate balance between transport downstream and thinning to

below the $1/3H$ observed threshold. Relatedly, from Eq. 2 effective viscosity is proportionally related to effective strain as

$$\eta \propto \dot{\epsilon}_e^{(1-n)/n} \tag{7}$$

or, if $n = 3$, as $\eta \propto \dot{\epsilon}_e^{-2/3}$, where $\dot{\epsilon}_e^2 = \frac{1}{2}\mathrm{tr}(\dot{\boldsymbol{\epsilon}}^2)$ is the effective strain rate. This creates another balance whereby increasing the effective strain (or effective stress through $\tau_e \propto \dot{\epsilon}_e^{1/n}$) reduces the effective viscosity, encouraging convection, but also increases ice-column disturbance, discouraging convection. While our experiments focus on along-flow slices, it is possible

that this may assist explanations regarding the presence of tall plumes with off-axis orientations just outside ice stream margins (Frank et al., 2022; Jansen et al., 2024) where surface velocity is higher. Here effective strain is greater, and hence effective viscosity reduced, but through-column horizontal shear is not excessive (rather, rotation can increase effective strain while not significantly disrupting plumes). Modelled upwards velocity rates may push through pRES measurement and location uncertainty making analysis of repeat radar surveys a feasible way to test if these plumes are actively expanding. Notably, the

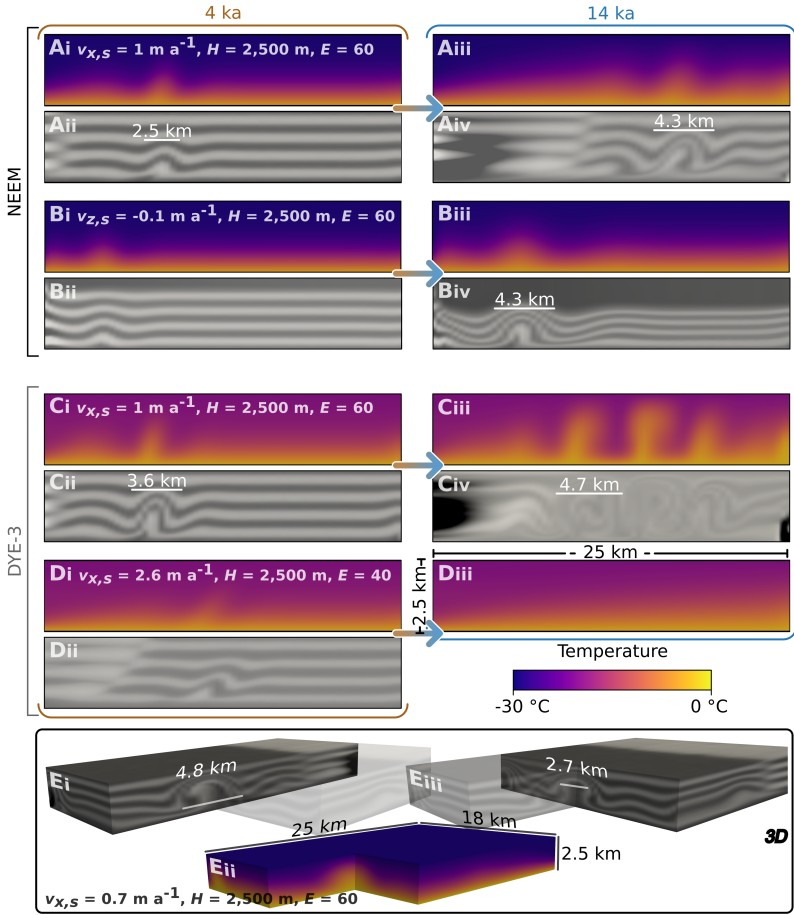

**Figure 6.** Snapshots of model behaviour at 4 and 14 kyr as marked in Fig. 4. Panels with plasma colormap display temperature, while the accompanying grayscale colormap in each instance shows disruption of initially flat horizontal layering used to simulate isochrones. Numerical diffusion leads to the pattern in **Civ** being slightly difficult to distinguish. The pattern corresponding to **Div** is unintelligible and therefore omitted. Each domain has a length of 25 km, a height of 2,500 m and a vertical exaggeration factor of 2. Panel **E** shows the 3D simulation run corresponding to the marking in 4F. Fig. A5 mirrors Fig. 2 but displays $v_x$ and $v_z$.

convection plumes generated in our 3D simulations (Fig. 6E) more closely resemble the geometry of observed folds (Fig. 2C, D), and we are not aware of another mechanism that is hypothesized to produce this unique type of geometry.

Such settings are rarer or absent in southern Greenland, where escape times from the central ice divide to the 2,000 m principal contour line are only a little over 10 kyr. The significant snowfall ($\geq 0.35$ m yr$^{-1}$) and hence downwards motion in south Greenland (Fig. 1E) also likely limits the possibility of convection in this region (Fig. 4J) and will have done for at least the past 9 kyr (MacGregor et al., 2016).

Considering existing hypotheses for the formation of the observed folds, basal freeze-on may be limited as a general explanation capable of explaining plume ubiquity in north Greenland (Bell et al., 2014; Leysinger Vieli et al., 2018) given the requirements for large volumes of basal water (Dow et al., 2018) in a region that is not likely to be pervasively thawed (Bons et al., 2018; Macgregor et al., 2022). Further, freeze-on may not explain the fairly consistent sizing and spacing (at ∼10 km, Fig. 2) of some north Greenland plumes, which contrast the much more spatially extensive freeze-on layers in East Antarctica (Bell et al., 2011). Travelling slippery spots (Wolovick et al., 2014) develop clearly in an controlled setup, but also require thawed bed areas in the same region to facilitate at least a degree of slip and further do not appear to align with the observation of a highly deformed basal layer beneath the plumes (Fig. 2C), which may be more consistent with high rates of basal ice deformation than basal sliding (Zhang et al., 2024b). Travelling slippery spots may also not be compatible with the 3D geometry of observed plumes (Figs. 2B, D) or with ice motion over a rough bed. Additionally, neither mechanism accounts for an apparent absence of $H > 1/3$ plumes in south Greenland. However, the basal thermal state may have been different ∼10 kyr ago and we do not rule out these two processes contributing to the onset of an initial perturbation or playing a role in their continued development. Our imposition of a no-slip, constant-temperature basal boundary also means that possible feedbacks between convective heat dispersal and basal sliding are not recognised. These may complicate plume geometry in a similar manner to that explored in Wolovick et al. (2014). Rheological contrasts (Dahl-Jensen et al., 2013) and convergence (Bons et al., 2016), as covered in Zhang et al. (2024b) may also interact with convection in ways not explored here. Last, geothermal heat flux decreases throughout our simulations towards ∼4 mW m$^{-1}$. This will have some bearing upon model outcome but as outlined in the Methods a linear temperature profile with constant geothermal heat flux presents its own misrepresentations. In any case, we still obtain stable max($v_z$) time series for the sustained regime, while the amplifying regime may or may not reach a steady state (Fig. 5). We highlight these possibilities to motivate further work on englacial plumes and more clearly determine to what degree it is necessary for convection to operate in concert with additional processes.

Is it possible that effective viscosity values lower than commonly assumed are feasible for the northern GrIS? Independent of convection being possibly the only feasible mechanism for large plume formation, we suggest that an affirmative answer may be appropriate. Large plumes are mostly found in areas with a relatively larger proportion of pre-Holocene ice (Fig. 1A). Beyond this observation fulfilling the requirement for relatively stable ice (condition (3)), older ice from the Last Glacial Period is consistently measured or inferred to be significantly less viscous than Holocene ice (Paterson, 1991; MacGregor et al., 2016; Bons et al., 2018; Law et al., 2021), as a result of fabric development and a higher impurity content from a drier and dustier ice age. Despite the importance of this softer ice for interpretation of the overall motion of the GrIS, very few direct measurements exist. Borehole closure rates from ice divides likely reflect stresses inconsistent with basal shearing, and such locations are

often explicitly selected for their lack of a history of extensive horizontal shear (Talalay and Hooke, 2007). To our knowledge, no laboratory measurements have been conducted on ice resembling that found within what we hypothesize to be englacial convection plumes. It may therefore be possible that basal ice in north Greenland is sufficiently soft as to permit convective plume formation. Alternatively, additional plume-forming processes operating in parallel may expand the effective viscosity range required for their formation. Similarly, parallelly-operating processes may also account for the presence of plumes in

regions of slightly greater velocity than the threshold indicated here. More complex modelling featuring additional processes and tests on field specimens presents the clearest opportunity to directly asses our hypothesis.

In situ rheology is also modulated by anisotropy, which is not included in our simulations. Zhang et al. (2024b) suggest an important role for anisotropy in the formation of large plume-like folds (their Fig. 4) as a result of direction-dependent softening due to directional alignment of the c-axis. Plume-forming motion will rotate initially bed-planar fabric such that

it also broadly aligns with the dominant shear direction in plume formation. However, anisotropy itself describes a stress-orientation dependent rheology, rather than a specific softening. The role of anisotropy in Zhang et al. (2024b) then comes in part from their implementation which allows the viscosity acting along the plane perpendicular to the $c$-axis maximum (denoted $\eta_2$) to decrease by a factor of three and fall below the $1 \times 10^{13}$ Pa lower limit set for the isotropic run (their Table S2). Such a decrease is sufficient to reach the effective basal viscosity values ($\sim 3 \times 10^{12}$ Pa s) in our $E = 40$ and $E = 60$

simulations, where local convection becomes increasingly viable. This is not to say that progressive anisotropic softening is not an important process here. In glacier settings, bulk viscosity will generally decrease as ice develops stronger crystallographic anisotropy, though the effect is stress-state dependent (Azuma, 1994), meaning such softening may also be a contributing factor for plume locations occurring at distance from ice divides. In our application, as in many others (Cuffey and Paterson, 2010), the enhancement factor operates as a simple parameterisation of anisotropic effects without recourse to a stress-orientation

dependent tensorial flow law. Consequently, our results provide a plausible estimate of how progressive anisotropic softening may affect $\eta_2$, which likely exerts the strongest rheological control on convection onset.

Ice is more accurately represented as a non-linear shear-thinning fluid in most situations, and may also be more non-linear than the linearised approximation of $n = 3$ implemented in this study, with growing evidence for $n = 4$ in some regions (Bons et al., 2018; Ranganathan and Minchew, 2024). Our use of Newtonian rheology allows us to test a broad parameter space

but may miss non-linear interactions caused by the plumes themselves which both increase and decrease effective strain rates and thereby influence the effective viscosity (Eq. 7). Increasing values of $n$ away from unity may increase the importance of these non-linear stress responses within the plumes; intuitively one may anticipate in a direction that more readily facilitates plume formation, though this will depend upon the appropriate values for $A_0$ and $Z$ and the resultant effective strain field. We emphasise, however, that rate-weakening in plumes is still expected to be small compared to the main coastward movement

of the ice sheet, which exerts a first order control over effective stress (Figs. A2, A5). In any case, we hope that our results closely isolate the effective rheological thresholds for ice-sheet convection, which permits a narrower starting point for future numerical models featuring more complex and computationally costly thermodynamics.

A lower effective viscosity of basal ice will significantly influence ice dynamics, similar to the influence of an increased flow exponent, $n$ (Bons et al., 2018; Zeitz et al., 2020; Ranganathan and Minchew, 2024). If ice-sheet models are initiated under

fixed assumptions of higher ice viscosity, then inversions for basal traction will overcompensate by producing unrealistically low basal traction values and bias the resulting projections (Berends et al., 2023). Convection-driven plumes also present a mechanism that draws warmer and lower-viscosity basal ice upwards – counteracted by the transport of higher-viscosity (colder) interior ice downwards. Exploring the possible implications of lower viscosity ice and convection-driven mixing – and their influence upon inferred basal traction – is therefore warranted to better quantify the errors that may be introduced

into predictive ice-sheet models. Finally, the relative lack of large plume observations in the Antarctic Ice Sheet, outside of the Gamburtsev Mountains, may simply result from colder temperatures there and hence higher viscosities in the upper ice column, which limit convection (Fortuin and Oerlemans, 1990; Bell et al., 2011; Cavitte et al., 2021; Sanderson et al., 2023), or from a sampling bias given the comparative paucity of radar-sounding observations in Antarctica (Bingham et al., 2024).

## 5   Conclusions

Our modelling indicates that, following an initial perturbation, local convection is possible within the Greenland Ice Sheet under conditions that are not unrealistically far from the existing consensus on ice rheology. This hypothesis could explain the observed spatial distribution of large plumes in Greenland, with surface velocity, accumulation rates and ice rheology exerting the strongest controls on convection viability and hence plume formation. A corollary of this result is that ice in northern Greenland may be softer than commonly assumed. Appropriately probing and then implementing these constraints

into ice-sheet models may help reduce compensatory errors and improve the accuracy of their future projections.

*Code and data availability.* The supporting data is available at https://doi.org/10.5281/zenodo.14892876 .

## Appendix A: Appendices

### A1  Rayleigh number

The Rayleigh number following Rayleigh (1916) is calculated as

$$Ra = \frac{H^3 \Delta T \beta g \rho}{\alpha \eta}, \tag{A1}$$

where $H$ (m) is the thickness of the fluid layer, $\Delta T$ (K) is the temperature difference between the surface and base, $\beta$ (K$^{-1}$) is the thermal expansion coefficient, $g$ (m s$^{-2}$) is the acceleration due to gravity, $\bar{\rho}$ is the base material density, and $\alpha$ (m$^2$ s$^{-1}$) is the thermal diffusivity. Previous attempts have been made to determine a critical Rayleigh number for non-Newtonian fluid layers (e.g. Ozoe and Churchill, 1972) but this becomes complicated by their dependence on the amplitude and shape of the disturbance initiating motion (Parmentier, 1978).

If we extend the basal viscosity (calculated at $5 \times 10^4$ Pa effective stress and $-2°$C) uniformly through a 2,500 m ice column with $\Delta T = 30$K following Fowler (2013), we obtain a Rayleigh number of 2805 for $E = 5$ (Fig. 3). Hughes (2012) and Fowler (2013) extend this approach in their arguments, with Fowler (2013) emphasizing that the lack of an initial thermal perturbation will prevent convection onset. However, as covered in the discussion, bedrock perturbations (e.g. Figs. 2B, C) or basal folding induced by other processes (Zhang et al., 2024b) can easily satisfy this challenge.

### A2  ISSM run

The ISSM (Ice-sheet and Sea-level System Model) run was completed following the setup of the UCI_JPL group featured in Goelzer et al. (2020) with a higher-order Stokes approximation and an ISSM Budd sliding relationship relating basal traction, $\tau_b$, to basal velocity, $v_b$, as

$$\tau_b = C^2 N^r v_b^s \tag{A2}$$

where $C$ is the traction coefficient, $r = \frac{q}{p}$, and $s = \frac{1}{p}$ where $q$ and $p$ are parameters both set to 1. Following an inversion procedure to calculate basal traction the effective pressure, $N$, is calculated as

$$N = \rho_i g H + \rho_w g b_z \tag{A3}$$

where $\rho_i$ and $\rho_w$ are the densities of ice and water respectively and $b_z$ is the position of the bed. $n$ in Eq. 1 was set to 3, and an initial approximation of ice rigidity is made based on initialised ice temperature (details in Larour et al., 2012). The model was run transiently for 0.25 years with a timestep of 0.01 a. The Matlab runscript can be found in the Open Research Section. This model was used only to give an indication of expected effective stresses within the GrIS and will reflect effective stress in

most standard ice-sheet modelling applications. We did not re-run the model with updated enhancement factors suggested in this paper, or a greater value of $n$, both of which may influence the effective stress and therefore effective viscosity. The reader is referred to Larour et al. (2012) for a more detailed model and ice physics description.

$\tau_e$ remains surprisingly uniform throughout the ice column, justifying our use of a constant value (and, deciding on the form of a variable $\tau_e$ profile with depth would present its own issues, hence we opt for the simplest approach for transparency). This can be understood in part through the relationship of $\dot{\epsilon}_e$ and $A$ with depth. We can set $\dot{\epsilon}_e = A\tau_e^n$ or $\tau_e = (\dot{\epsilon}_e/A)^{1/n}$ and then $\ln(\tau_e) = 1/n(\ln(\dot{\epsilon}_e) - \ln(A))$ which gives:

$$\frac{d}{dz}\ln(\tau_e) = \frac{1}{n}\left(\frac{d}{dz}\ln(\dot{\epsilon}_e) - \frac{d}{dz}\ln(A)\right). \tag{A4}$$

This means that the variation in $\tau_e$ with depth arises from the difference between the log of $\dot{\epsilon}_e$ and $A$, but as both are increasing with depth in our setup the variation is not that large. For example, both $\dot{\epsilon}_e$ and $A$ may be expected to vary by one order of magnitude between -15 and -1$^o$C and half and full depth, respectively.

## A3 Model run times

Each set of roughly 15 values each over a 20 ka period in 2D takes around 36 hr on 8 2-GHz CPUs. Running one 3D simulation for 14 ka takes around 84 hr on 48 2-GHz CPUs.

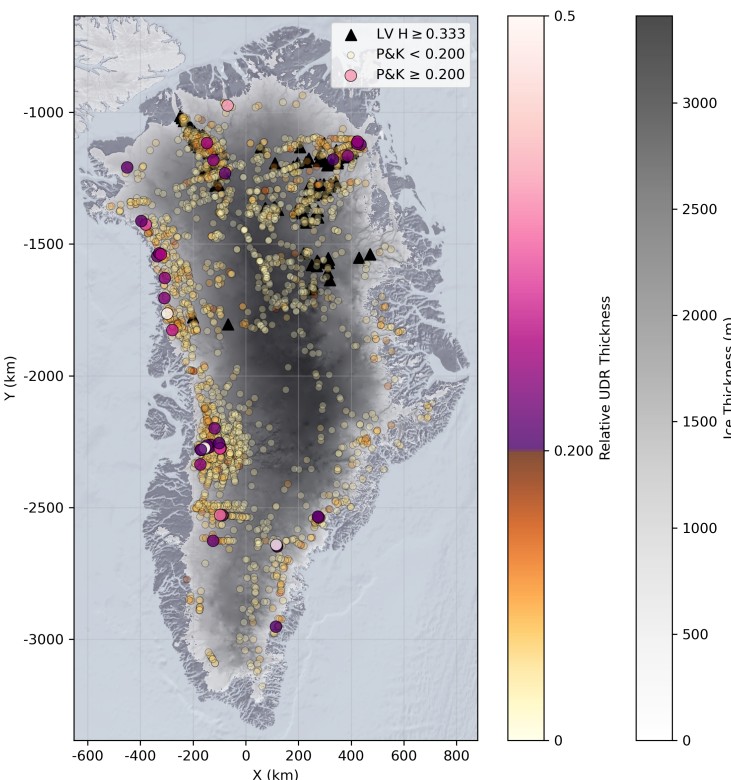

**Figure A1.** Map showing automated mapping of units of disrupted radiostratigraphy (UDRs) from Panton and Karlsson (2015) (circles) and traced plumes from Leysinger Vieli et al. (2018) (triangles). Panton and Karlsson (2015) UDRs are filtered to only include those detected where ice thickness exceeds 1 km. Relative UDR thickness is calculated using BedMachine v5 (Morlighem et al., 2022).

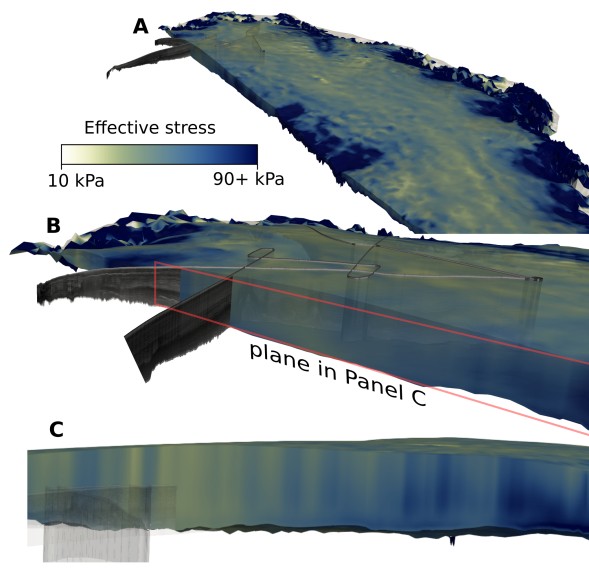

**Figure A2.** Effective stress obtained from ISSM run with the same radar transects shown in Fig. 2.

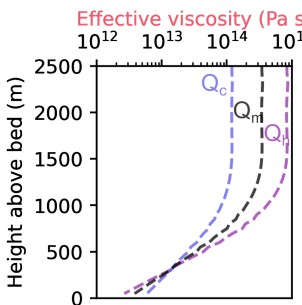

**Figure A3.** Effect of different $Q$ values on effective viscosity for $E = 40$ and the NEEM temperature profile. $Q_c = 6 \times 10^4$ is for $T < -10°\text{C}$, $Q_h = 11.5 \times 10^4$ is for $T \geq -10°\text{C}$, and $Q_m = 9 \times 10^4$ is the midpoint used in this study.

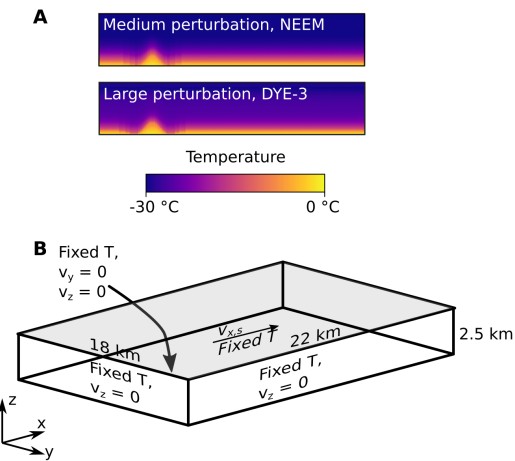

**Figure A4.** A. Medium and large temperature perturbations, at the same scale as Fig. 3. B. Boundary setup for 3D simulations.

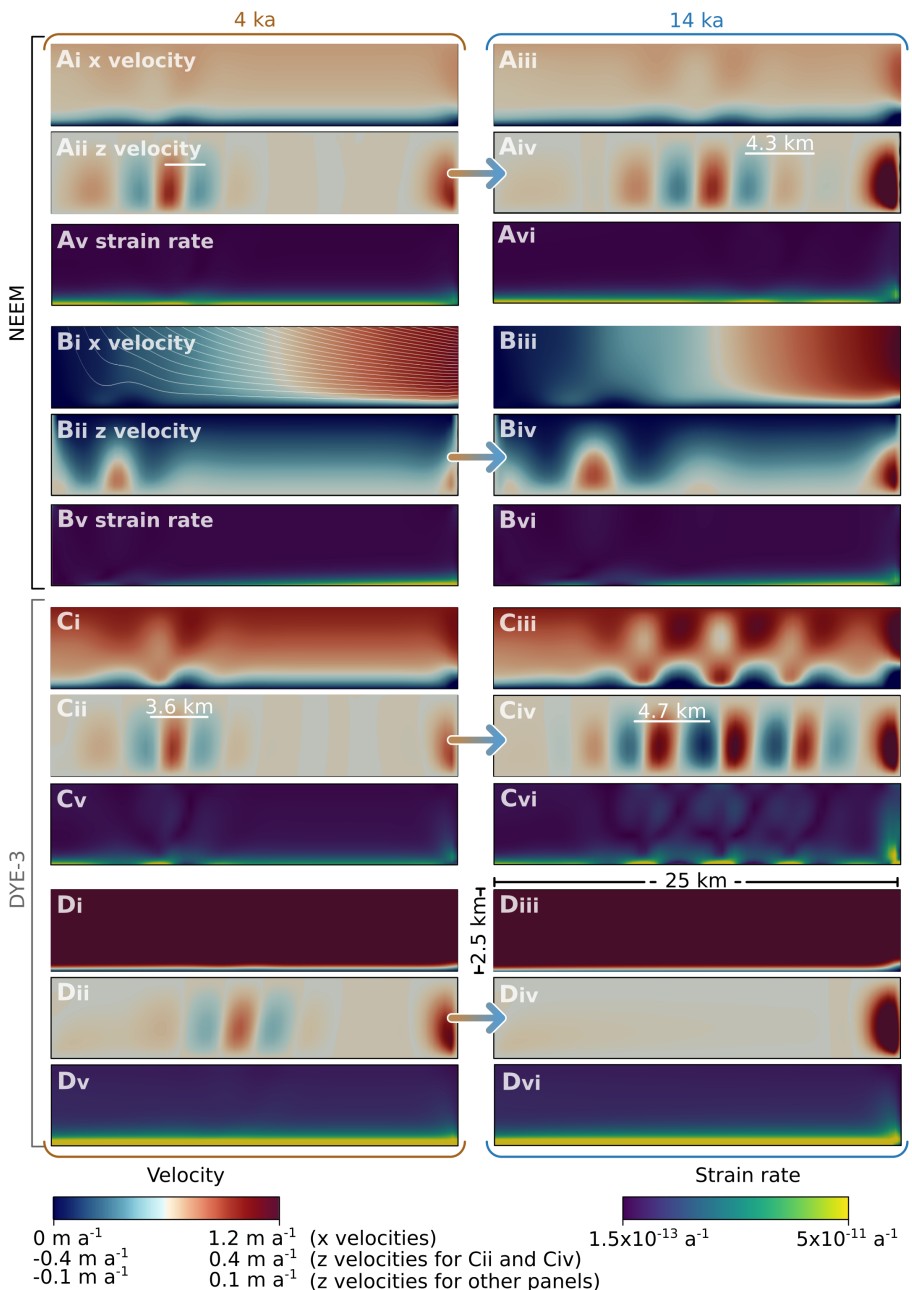

**Figure A5.** As for Fig. 4 but showing x and z velocity components and effective strain rates. Panel Bi additionally shows flow lines originating from the surface at a spacing of 625 m. For Panel Avi, the effective strain rate in the mid column of $1\times10^{-12}$ may increase by 50% or decrease by 20% due to convection. For panel Cvi the effective strain in the mid column of $3.5\times10^{-12}$ may increase by 100% or decrease by 60% due to convection.

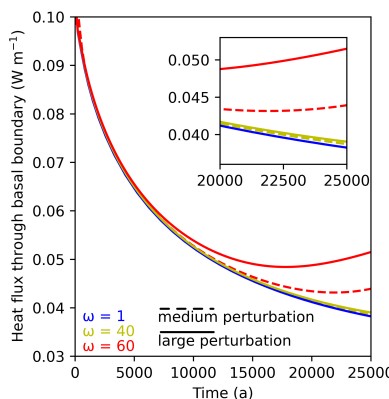

**Figure A6.** Domain-averaged heat flux through the basal boundary for simulations run with the NEEM temperature profile, an ice thickness of 2,500 m, and 0 surface velocity. The dashed line refers to the medium perturbation, while the solid line refers to the large perturbation. A detail panel with the same axis is shown as an inset. The initial decrease in heat flux in all runs is a result of the basal temperature gradient (Fig. 3) reducing over time, before convection increases the heat flux again in simulations where the enhancement factor is 40 and 60. The medium perturbation results in a lower heat flux at a given time after ∼10 ka as the large perturbation essentially gives the convection state a head start

**Table A1.** Set parameters

| Parameter | Value |
| --- | --- |
| Activation energy, $Q$ | $9 \times 10^4$ (kJ mol$^{-1}$) |
| Ideal gas constant, $R$ | 8.314 J K$^{-1}$ mol$^{-1}$ |
| Creep prefactor, $A_0$ | $3.5 \times 10^{-25}$ Pa s |
| Flow exponent, $n$ | 3 |
| Specific heat capacity, $C_p$ | 270 J kg$^{-1}$ K$^{-1}$ |
| Reference density, $\bar{\rho}$ | 917 kg m$^{-3}$ |
| Thermal expansion coefficient, $\beta$ | 380 K$^{-1}$ |
| Thermal conductivity, $\kappa$ | 370 W m$^{-1}$ K$^{-1}$ |
| Thermal diffusivity, $\alpha$ | $3.5 \times 10^{-6}$ m$^2$ s$^{-1}$ |

**Table A2.** Run setups corresponding to panels in Fig. 4. Format of e.g. 25:5:100 indicates steps from 25 to 100 in spacing increments of 5. Run format of e.g. B-3Di corresponds to the first 3D run within the parameter space of B. B-3Div is equivalent to F-3Dii.

| Run | $E$ | $v_{x,s}$ | $H$ | $v_{z,s}$ | T profile | Perturbation |
|---|---|---|---|---|---|---|
| A | 25:5:100 | 1 | 2,500 | 0 | DYE-3 | large |
| B | 25:5:100 | 1 | 2,500 | 0 | NEEM | large |
| C | 25:5:100 | 1 | 2,500 | 0 | NEEM | medium |
| D | 40 | 0:0.2:3 | 2,500 | 0 | DYE-3 | large |
| E | 40 | 0:0.2:3 | 2,500 | 0 | NEEM | large |
| F | 60 | 0:0.2:3 | 2,500 | 0 | NEEM | large |
| G | 40 | 1 | 1,800:100:3,500 | 0 | DYE-3 | large |
| H | 40 | 1 | 1,800:100:3,500 | 0 | NEEM | large |
| I | 60 | 1 | 1,800:100:3,500 | 0 | NEEM | large |
| J | 40 | N/A | 2,500 | 0.02:0.02:0.12, 0.15:0.05:0.4 | DYE-3 | large |
| K | 40 | N/A | 2,500 | 0.02:0.02:0.12, 0.15:0.05:0.4 | NEEM | large |
| L | 60 | N/A | 2,500 | 0.02:0.02:0.12, 0.15:0.05:0.4 | NEEM | large |
| B-3Di | 30 | 1 | 2,500 | 0 | NEEM | 3D |
| B-3Dii | 40 | 1 | 2,500 | 0 | NEEM | 3D |
| B-3Diii | 50 | 1 | 2,500 | 0 | NEEM | 3D |
| B-3Div = F-3Dii | 60 | 1 | 2,500 | 0 | NEEM | 3D |
| F-3Di | 60 | 0.7 | 2,500 | 0 | NEEM | 3D |
| F-3Diii | 60 | 1.3 | 2,500 | 0 | NEEM | 3D |

*Author contributions.* RL designed and completed the experimental setup, produced the figures, and wrote the initial manuscript with input from all co-authors. JAM supported with radar data visualisation. CMG assisted with ASPECT model setup.

*Competing interests.* No competing interests are present.

*Acknowledgements.* RL, AB, and PV acknowledge funding from Norges Forskingsråd (SINERGIS project, Norwegian Research Council Grant 314614). CMG is supported by the UK Science and Technology Facilities Council [grant number ST/W000903/1]. JM acknowledges support from the NASA Cryospheric Sciences Program. Thanks to Gwendolyn Leysinger Vieli for the background information on the plume locations.

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
