# Peer review of "Exploring the conditions conducive to convection within the Greenland Ice Sheet"

_EGUsphere, 2025_

## Referee Comment (RC1)

**Review of, "Exploring the conditions conducive to convection within the Greenland Ice Sheet", by Law et al.**

Review by Michael Wolovick

**Summary**

In this paper, Law and coauthors use geodynamic modeling techniques to explore the possibility that local thermal convection may occur in the Greenland Ice Sheet. Thermal convection has long been hypothesized for continental ice sheets, mostly by Terry Hughes, yet it has never seen observational confirmation or enjoyed widespread acceptance in the glaciological community. However, over the past decade or so a growing literature has developed around the observation of large plume-like folds in northern Greenland. There is no consensus on the formation mechanism of these plumes, with proposed causal mechanisms including basal freeze-on, traveling slippery patches, rheological contrasts in the ice column, and cross-flow convergence. This paper adds to the literature on these enigmatic plumes by proposing that they may have formed through buoyant thermal convection, thus posthumously vindicating Terry Hughes' long and lonely crusade to gain support among glaciologists for the existence of thermal convection in ice sheets.

The bulk of this paper is spent describing and analyzing the results of applying a geodynamic model to Greenland-like conditions. The authors' model results indicate that thermal convection may in fact be possible under the conditions that prevail in north Greenland. They find that the development of convection is inhibited by higher surface accumulation rates and faster flow speeds, explaining why plumes are not observed in south Greenland. Their model requires ice viscosity significantly lower than standard rheological assumptions, which, if true, would have important implications for the modeling of ice sheet dynamics and for the results of basal traction inversions.

**Major Comments**

This paper is clearly appropriate for The Cryosphere. It represents both an important addition to the literature around a relatively new observational mystery (the large englacial folds) and a dramatic coda to a very old theoretical debate (thermal convection). The paper is well written and argued. Some of the figures need work in order to better display the authors' results. In particular, they seem to have crammed their entire parameter space exploration into Figure 3, which is quite busy and difficult to understand. However, they only have four figures total in the main text, so they have plenty of room to split this information into an additional figure to aid comprehension. Their model has some issues, which I discuss next, that potentially undercut their conclusion that rheology must be softer than commonly assumed. However, their model also has some strengths relative to typical ice sheet models, and their overall conclusion that thermal convection can explain the observed plumes is supported by the model results presented. My concerns about the simplifications made in their model mainly fall under the heading of, "all models are wrong, but some models are useful", and my concerns can be addressed through changes in the discussion or conclusions text, rather than a redo of the modeling itself. My recommendation is to publish with minor revisions. I now go on to discuss my main concerns with their model setup.

The authors use a geodynamic model rather than a classic ice flow model. This has some advantages that make it better for simulating convection, such as the inclusion of a buoyancy term and the use of the Full Stokes equations. However, it does mean that, because their model was not originally designed for glaciological applications, their implementation of boundary conditions and ice rheology are somewhat problematic.

For rheology, they used a linear Newtonian rheology for ice, derived from the non-Newtonian rheology by assuming a constant effective stress ($\tau_e$=50 kPa) throughout the domain. They include the temperature dependence of rheology, but not the stress (or equivalently, strain rate)

dependence. They justify this omission by claiming that the strain rates associated with convection are much lower than the background strain rates associated with horizontal ice flow, but the background strain rates should be highly concentrated near the bed, while they have used a constant 50 kPa value for $\tau_e$ rather than a vertically variable one. It is not necessarily clear to me that the strain rates associated with convection will be lower than the background strain rates in the mid-column, and the authors have not shown this comparison to justify their assumption. While I do not believe that the use of a linearized rheology undercuts their conclusion that thermal convection is possible for the conditions that prevail in north Greenland, it does undercut their conclusion that the enhancement factor must be much larger than typically used in ice flow models. The authors have concluded from their model results that E must be an order of magnitude larger than the typical value, but their results could just as easily be interpreted to mean that they used a value of $\tau_e$ that was too small. Additionally, it is not necessarily clear that the particular thresholds for convection that they found when varying E would hold up if they had used either a vertically variable $\tau_e$ or a fully nonlinear rheology.

For the boundary conditions on the upper surface, they used Dirichlet conditions to impose both horizontal flow and surface accumulation, rather than having a stress-free free surface like the real ice sheet. This could have been more problematic but did not end up being a huge issue because they included the temperature dependence of rheology, so shear was concentrated near the bed anyway. A bigger issue is the lower surface: they did not state what boundary conditions they used for temperature, but I was able to infer from other parts of their model setup and results that they used a Dirichlet condition with basal temperature set to -2°C, which is a reasonable approximation of the pressure melting point under 2 km of ice. It is likely that they did this because the classic convection problem in fluid mechanics or geodynamics involves a fixed ΔT across a specific layer thickness. However, in ice sheets, the basal boundary condition is typically Neumann (gradient determined by geothermal heat flow) up until the point that the basal temperature reaches the melting point, when it switches to Dirichlet. Thus, the authors' use of a Dirichlet condition for temperature at the ice base implies that the ice base is wet, but this contradicts statements in the paper that the authors believe that the bed is frozen where the plumes are observed, and also contradicts the authors' use of a no-slip condition at the ice base.

The presence of sliding at the ice base does not necessarily undercut the authors' conclusion that thermal convection is possible; if anything, since they found that vertical shear suppresses the development of convection, allowing basal slip may actually broaden the parameter range over which convection is possible. However, the presence of sliding may undercut their rheological conclusions, where they argue that sliding is less extensive than existing inversions have found. Also, if the basal temperature is tied to the melting point, then the temporal and spatial variations in conductive heat flow that they found (Figures 4 and S7) should be associated with changes in the melting or freezing rate, which should impact the availability of basal water and the basal traction. Those changes in the basal boundary condition should, in turn, influence the ice flow field. Thus, we would expect thermal convection to interact with both the basal freeze-on and traveling slippery patches mechanisms. In order to truly model convection alone, the authors would need to use Neumann conditions for temperature at the ice base, and choose a parameter range where they were sure that the base would never warm to the melting point. A model of thermal convection in ice sheets with a warm bed but no sliding and no variations in basal melting or freezing is necessarily omitting some pretty important feedbacks.

To be clear, I do not think that there is anything wrong with writing a paper focused on convection alone, especially considering convection's contentious history in glaciology, and considering the fact that the original papers on basal freeze-on and traveling slippery patches (which I was involved in as either first author or coauthor) completely ignored thermal buoyancy. However, when discussing results and drawing conclusions, it is very important to make note of what processes were omitted from the model, and to think about how those processes might affect the results. In reality all processes are coupled together, and we cannot think of convection as being truly independent of basal freeze-on or traveling slippery patches (or, for that matter, from

anisotropy and cross-flow convergence).  In the real world, the basal boundary condition is not fixed and immobile; rather, the variations in the englacial temperature field seen in the authors' model will produce spatial and temporal changes in the conductive heat flux at the ice base, which produces changes in the melting or freezing rate, changing the availability of basal water and basal traction, which finally feeds back on the original englacial flow field produced by convection in the first place.  Even if the background temperature field is initially cold-based, the uplift underneath a rising buoyant plume is likely to have a lower conductive gradient than its surroundings, warming the bed locally and potentially producing a local patch of basal melt. Do these feedbacks act in a way that amplifies convection, or suppresses it?  Will they broaden the parameter range over which convection is possible, narrow it, or merely shift it?

As I said previously, I think that my criticisms of the authors' model setup mostly fall under the old saying, "all models are wrong, but some models are useful".  The authors used a model that is well-designed for simulating convection and poorly designed for other things.  In most geodynamics problems there is no need to simulate a dynamic basal boundary that switches modes in response to changing conditions within the model domain; thus, the authors' model can't include subglacial hydrology.  This doesn't mean that the authors' results have no value.  On the contrary, they have showed, at long last, that thermal convection is possible in ice sheets, and they have connected this model result with an enigmatic set of observations that could plausibly be caused by convection.  This is an important result and worthy of publication.  My main concern is that the discussion and conclusions sections need to be more forthright about the limitations of using a geodynamics model for glaciological problems, and these sections should also discuss potential feedbacks between thermal convection and other mechanisms that might contribute to the observed folds, especially mechanisms that are sensitive to the thermal state of the ice base.  Finally, I think that the authors' conclusion about the likely value of the enhancement factor E should be given more caveats.  Given the simplifications that they made to ice rheology in their model setup, and given that they omitted feedbacks that could potentially change the parameter range over which convection is possible, I am not convinced that we can necessarily use the observed plumes to infer that ice is softer than commonly believed.  The authors are free to keep arguing that, of course, but I think some caveats are necessary for that particular conclusion.

**Minor Comments**

Note that my copy of the manuscript does not have line numbers.  I will try to place my comments by giving the section of the paper and a quote.

Introduction:

"At first glance, an entirely separate problem is the nature of the formation of large (>1/3 the local ice thickness) englacial plumes found by tracing reflections of equal age in radargrams (i.e., isochrones; Figs. 1A, 2, S1, CReSIS (2013))"
Add a reference to Bell et al. (2014) here.  You have a reference to Bell elsewhere in the paper, but Bell et al. (2014) was also the first to describe the large plume-like reflectors in northern Greenland and they should be cited here, even if you don't agree with their interpretation of the reflectors as originating with freeze-on.

"Such plumes have previously been hypothesized to result from basal freeze on (Leysinger Vieli et al., 2018), or traveling basal slippery spots (Wolovick et al., 2014), which both require an at least temporarily thawed bed."
The end of this sentence is awkwardly phrased.  It also misses the issue of spatial variability, which is important to both mechanisms in addition to temporal variability.  Maybe try, "...which both require that the bed be at least locally or temporarily thawed".

"Both authors approach convection analytically only, by estimating a Rayleigh number Ra, the dimensionless ratio of heat transfer via upwards mass transport (i.e. convection) vs. thermal conduction (Rayleigh, 1916)...The formulation of thermal diffusion in Ra does not capture dynamical effects important in ice sheet flow, such as horizontal shearing; and the critical Ra is itself tied to the particular boundary conditions and the initial perturbation geometry"

Additionally, this analytic approach misses the fact that heat is *already* being advected by mass transport within the ice sheet- vertical thinning and subsidence associated with surface accumulation is pushing colder ice down. Any upwards convection needs to fight against this background state of subsidence, as you found in your model results later.

Materials and Methods:

"Surface mass balance is set to zero, i.e., no snowfall or surface melting ($v_{z,s} = 0$ at the surface boundary condition … surface shearing velocity $v_{x,s}$ is uniform across the domain's top surface"

Does this mean that you apply Dirichlet conditions to velocity at the upper surface of the domain, rather than stress-free (Neumann) conditions? Does this also mean that your upper surface is a rigid lid, rather than a free surface? The correct boundary conditions are Neumann, and the correct way to induce horizontal flow is to have a slope in the free surface that drives ice flow downstream, and the consistent way to induce vertical flow is to have the gradient of this induced horizontal flow have an along-flow gradient that balances the accumulation rate. I suspect that you have done things this way because your geodynamic model was not built to study ice dynamics, and so the steady state that I just described is beyond its ability to compute.

I don't think that this necessarily invalidates your results, mostly because Figure S5 shows that horizontal velocity is mostly uniform in the upper column, with shear mostly concentrated in the lower column. If your model had produced too much shear in the upper column, I would suggest throwing the whole thing out. However, you need to explicitly state that you use Dirichlet conditions rather than Neumann conditions at the upper surface, and you need to have some text justifying this choice and explaining what impact (if any) it has on your results.

In addition, you do not state what boundary conditions you use for temperature on the lower surface, but because heat flow across the lower boundary varies in time (Figure S7), I infer that you also used a Dirichlet condition for temperature, with a value of -2°C (roughly accurate for the melting point underneath 2 km of ice). I discussed my issues with this at greater length in the Major Comments section above; for here, my main comment is that the basal boundary condition needs to be stated explicitly.

"A Newtonian rheology is appropriate here as strain rates due to convection are small compared to those from background ice flow (Fig. S5)"

Figure S5 does not show strain rates, it shows velocities. I would like to see a figure for strain rates to support this argument. In particular, I am curious about whether the strain rates due to convection are smaller than the background strain rates in the mid-column, not just near the bed. This is potentially an important caveat to your results and an important limitation of trying to use a Newtonian rheology to model a non-Newtonian material.

"Prescribing $\tau_e$ is also necessary as a full ice-sheet stress state can not be accurately replicated in a simplified along-flow slice"

Especially when you don't have a true free surface and instead produce horizontal flow by imposing a Dirichlet condition at the top of your domain.

"We set $F_{int} = 0$, thereby ignoring adiabatic heating and neglecting strain heating, to prevent simulations with greater $v_{x,s}$ and hence greater strain heating from evolving a different rheology along flow."

This is a decent first approximation for the slow-flowing areas of the ice sheet, but it is potentially an important limitation. At a velocity of 10 m/yr (appropriate for many of the observed plumes) and a driving stress of 100 kPa (a good ballpark number for ice sheet stresses in general), the integrated shear heating is 32 mW/m$^2$, or about half of the geothermal heat flow. This could potentially play an important role in warming and softening the basal ice, and also could contribute to convection by providing additional thermal energy in the lower ice column that needs to escape to the surface. At a velocity of 1 m/yr that would be reduced by an order of magnitude, but as I said, some of the plumes are observed where velocities are about 10 m/yr or even higher.

"We apply a transformation, T2 = ((T1+Ta)/Tb)(Tb +Ta ) where T1 is the original temperature profile, Tb is the basal temperature and Ta is an adjustment term used to raise the basal temperature to −2o C. The temperature profiles are stretched and compressed when adapted to the range of ice thicknesses."

Two issues: 1) does this transformation leave the surface temperature unchanged, or are you also changing the surface temperature? 2) More importantly, this temperature profile is not going to be in steady state with the enforced accumulation rate and ice thickness in your model. Your Figure S7 shows a pretty dramatic initial adjustment of the domain-average basal conductive heat flow. That could be because your initial temperature is not in steady state with the downwards advection that is actually in your model. Why did you use this approach instead of computing a steady state vertical temperature field for your given accumulation rate and ice thickness? It is true that you don't necessarily know the shape function for vertical velocity until you actually run your model, but you can still get much closer to a steady state by applying a simple approximation (like a Nye model) rather than simply scaling the DYE-3 or NEEM profiles, which come from particular locations where the accumulation rate and ice thickness do not necessarily match the values used in your particular experiments.

Figures:

Figure 1.
  Notes: 1) the descriptions for subplots b and c are swapped. 2) It would probably be good to reproduce the observed plume locations in every plot, not just a. 3) A shape factor of 0.8 is appropriate for n=3 rheology and constant temperature. With a more realistic temperature structure, shear should be more concentrated near the bed and the shape factor should be closer to 1. 4) Why evaluate effective stress at 5/14 depth? Where does this number come from? 5) Subplot e might be better on a log scale, since we are mostly interested in areas where the accumulation rate is quite low. Either that or just reduce the maximum of the color scale.

Figure 2.
  These echograms are way too dark, at least on my screen. It is very hard to see anything. You should adjust the color limits to improve visibility. The oblique view is also pretty hard to make sense of. Why do you display the entire ice sheet, instead of just zooming in on the region of interest?

Figure 3.
  There is a lot going on in this figure and it is hard to interpret. I would recommend splitting this figure into two figures. For one thing, the use of lines when you have timeseries data, but the lines don't actually represent progress over time, is very confusing. I would recommend that one figure be timeseries (ie, the x-axis should be time, with color representing some other parameter) so that the reader gets a sense of how the model evolves over time. Then another figure could focus on the parameter space exploration by showing 2D contour plots at various cross-sections through your 4D parameter space (the 4 dimensions being enhancement factor, surface speed, height, and snowfall). The second figure should not have any time dependence in it, you should just choose a

single metric to quantify the strength of convection (based on the final paragraph of the methods section this would either be the max vz at 20 ka or the change in max vz between 4 ka and 20 ka). Thus, the first figure gives the reader a sense of model evolution for a handful of representative parameter values, while the second figure shows the reader how the strength of convection varies as a systematic function of parameter space. But as it is now, Figure 3 tries to show both an exploration of parameter space and evolution through time, and the result is that there is simply too much going on in one figure.

Figure 4. "a vertical enhancement factor of 2"
        I think you mean vertical exaggeration? Vertical enhancement factor invited confusion with the rheological enhancement factor E. It might also be a good idea to adjust the color scale for the stratigraphy figures so that the layers have more contrast.

Discussion:

"(2) Total horizontal shear through the column must be less than around 1 m yr $^{-1}$"
        Many of the plumes (especially upstream of Petermann Glacier) are in ice flowing faster than this.

"(4) the enhancement factor must exceed around 45-75."
        See my major comments about the need for caveats around your rheological conclusions.

"condition (2) is likely satisfied by low surface velocities throughout northern central Greenland"
        I don't know about that. The region with velocity less than 1 m/yr is actually quite small. The region below the 10 m/yr contour is bigger, but still doesn't include many of the observed plumes. Later you suggest that the plumes may have formed further inland before being advected to their present locations, but it is worth emphasizing that the region within the 1 m/yr contour in Figure 1 is actually tiny, and it is associated with escape times of many tens of thousands of years. You also suggest that the plumes may have formed before the Holocene, when accumulation rates were lower, ice was thicker, and (presumably) flow was slower. However, any plume that is being passively advected by ice flow will also be shrunk substantially by vertical thinning: the characteristic vertical strain rate associated with surface accumulation is a/D, where a is surface accumulation and D is ice thickness, and for values of 10 cm/a and 3 km (being generous to plume survival with both parameters!), that is a strain rate of about $3 \times 10^{-5}$ a$^{-1}$, or a characteristic thinning timescale of 30 ka. Thus we would expect a plume that initially formed near the ice divide where flow was around 1 m/yr and then took several tens of thousands of years to be advected to its observed position would have been reduced from its original size by a factor of 1/e. Since the observed plumes are already a third to a half of ice thickness, that is clearly impossible. Less generous assumptions about accumulation rate and ice thickness can easily lead to a thinning timescale shorter than the Holocene.
        In addition, arguing that the plumes originally formed where ice flow is less than 1m/yr begs the question, "why don't we observe any plumes within the 1 m/yr contour now?". I suppose it is possible that the plumes formed during the LGM and then plume formation stopped during the Holocene, however, the Holocene stratigraphy above the plumes is deflected upward, which implies that they have been active during the Holocene. In my opinion, it is much more likely that the plumes formed close to their present-day locations (and that some of them are still forming!), where velocities are more likely to be around 10 m/yr rather than 1 m/yr. But in any event, you need to be explicit here that you are making a prediction: your argument implies that the plumes are currently being passively advected by ice flow, not actively forming. This prediction can be checked through the use of pRES data to measure vertical velocities within the ice column.
        I do not think that all of this is necessarily fatal to convection as a formation mechanism, since (as I discussed in my major comments above) the presence of basal slip may reduce the

vertical shear that tends to suppress convection, and also coupling between englacial convection and subglacial hydrology may modify the parameter range over which plumes form.  But in any event, I think that the discussion needs to spend more time dealing with a few facts:  1) all of the plumes are observed where ice flow is faster than 1 m/yr, in some cases an order of magnitude or more faster, and none are observed where flow is slower than 1 m/yr; 2) escape times from the 1 m/yr contour are substantially larger than plume formation times; 3) passively advected plumes should shrink over time in response to surface accumulation and vertical thinning; and 4) Holocene layers above the plumes are deflected upwards.  How do you square these facts with your model result indicating that convection requires velocities less than 1 m/yr, and your argument that the plumes initially formed during the LGM and are only being passively advected during the Holocene?  Or, if you think that the plumes are being actively formed during the Holocene, then why don't we observe any near the divide, and why do we only observe them where ice is flowing faster than your model says should be possible?

"Traveling slippery spots (Wolovick et al., 2014) develop clearly in an controlled setup, but also require thawed bed areas in the same region…"
        True, but I don't see why that's a problem.  There's plenty of uncertainty in the basal temperature and its not unreasonable to postulate at least local areas of thawed bed in the interior of Greenland.  Plus, your model setup has the basal temperature tied to the pressure melting point too.

"…and further do not appear to align with the observation of a highly deformed basal layer beneath the plumes (Fig. 2C), which may be more consistent with high rates of basal ice deformation than basal sliding (Y. Zhang et al., 2024)."
        The traveling slippery spots mechanism does not require 100% basal slip in the slippery patches, just a decent contrast in slip percentage between the slippery and the sticky patches.  Wolovick and Creyts (2016) goes into more detail about this, and also shows that the overturned parts of the folds are more likely to be found over the sticky parts of the traveling stick-slip trains anyway (DOI:10.1002/2015JF003698).

"we cannot rule out these two processes contributing to the onset of an initial perturbation."
        That's a good start, but I want to see greater discussion of the ways that convection can interact with other proposed mechanisms.  Not only traveling slippery spots and basal freeze-on, but the explanations based on rheological contrasts (DOI:10.1038/nature11789) and cross-flow convergence (DOI:10.1038/ncomms11427) too.  I realize that you do talk about anisotropy later in the discussion, but mostly in the context of whether a softer rheology is reasonable, not in terms of formation mechanisms for the plumes.

"We emphasize, however, that rate-weakening in plumes is still anticipated to be small compared to the main coastward movement of the ice sheet which exerts a first order control over effective stress (Figs. 1D, S1, S3)."
        First of all, I don't really see how Figs S1 and S3 address this question.  Second of all, as I discussed in my major comments, I buy the argument that the effective stress is dominated by the background flow of the ice near the bed, but I am not convinced that this is the case in the mid-column.  I would like to see a supplemental figure showing strain rates in the model, in addition to Fig S5 which shows the components of velocity.  The plumes are a relatively short-wavelength feature with vertical flow rates quite a bit larger than the vertical flow rates due to accumulation rate, so I strongly suspect that they are the dominant component of the strain rate tensor in the mid-column.

Appendix A1:

"If we extend the basal viscosity (calculated at $5\times10^4$ Pa effective stress and $-2°$C) uniformly through a 2,500 m ice column"

  This is a questionable assumption. It would be more accurate to use an effective viscosity appropriate for the mid-column, or at least the mid-lower column (for instance, one quarter or one third of the ice thickness).

Appendix A2:

"an initial approximation of ice rigidity is made based on ice temperature."

  Ice temperature from where?

Figure S2:

  This figure needs some work to make it a better visualization. It is way too dark and I really get no information from it at all.

Figure S3:

  As I mentioned in my major comments, using a single vertically constant value for effective stress is going to underestimate the vertical gradient in viscosity. Also, the caption should use "vertical exaggeration", not "vertical enhancement".

Figure S5:

  As I mentioned elsewhere, I want to see an additional figure showing the strain rate components (and overall magnitude) in addition to the velocity components.

Figure S7:

  This shows the average basal heat flux across the whole model domain, right? Your model has substantial local variation in the temperature structure, which should also produce local variability in the basal heat flux. It would be good if the caption explicitly stated that this is a spatial average of heat flux.

  It is also worth noting two things here: 1) the initial steep decrease in basal heat flow is likely because your initial temperature field is not in equilibrium with your imposed boundary conditions. If you started with a steady 1D thermal profile (or at least, a reasonable approximation of a steady 1D profile), then this initial adjustment would be much smaller. 2) You have imposed a constant basal temperature because that is standard in convection modeling. However, an ice sheet bed can only be held at a constant temperature if there is water present. If the basal heat flux increases because of convection, and there is no water present, then the basal temperature will drop, thus weakening the convection. Alternately, if there is water present, then the increase in heat flow will lead to freeze-on. If sufficient water supply is available (an important limitation), then the extra heat flow from convection will be provided by the latent heat of freezing water.

---

## Referee Comment (RC2)

**Review of, "Exploring the conditions conducive to convection within the Greenland Ice Sheet", by Law et al.**

Large plume-like folds are important features observed in the Greenland Ice Sheet, with significant implications for reconstructing past ice dynamics and projecting future ice-sheet mass balance. This paper investigates the formation of these plume-like structures. It is exciting to see the authors apply the well-established geodynamic modeling software ASPECT—commonly used in mantle plume studies—to test the hypothesis that thermal convection can produce such features. The modeling results are robust, as the authors present a comprehensive set of simulations exploring a wide range of enhancement factors, surface velocities, ice thicknesses, snowfall rates, temperature profiles, and model dimensions. Some of the simulated structures closely resemble those observed in radar profiles. The study finds that surface velocity, accumulation rate, and ice rheology exert the strongest controls on plume formation, which can also explain the spatial distribution of plume-like folds across northern and southern Greenland. Overall, this is a well-executed study, particularly valuable for improving our understanding of ice rheology in inland ice-sheet regions, where the ice may initially be frozen on the ground.

**General comments**

The model results clearly demonstrate that thermal convection can generate plume-like folds. I find the numerical results convincing, especially as they align with findings from my own modeling work on buoyancy effects (Y. Zhang et al., 2024). I also conducted tests using lower viscosities that show a more pronounced buoyancy-driven response. I'm excited to see this paper, as it represents a significant step forward in systematically constraining the controlling factors on convection and thoroughly analyzing the formation of plume-like structures. However, I have several concerns when relating the model results to the realistic world:

Ice rheology. The authors use a Newtonian and isotropic rheology. While the inclusion of an enhancement factor can approximate some aspects of natural ice behavior, this rheology may not be widely accepted. It would be good to include a paragraph at the end of Discussion section to carefully discuss the limitations of this ice rheology to the results. Actually I think a nonlinear and anisotropic rheology is likely to enhance convection, but am not sure how much initial perturbation would be required to trigger convection if the flow law is changed. It would also be helpful to explicitly state the range of effective viscosities in the convection models and, if possible, include representative viscosity maps. It is important to know the viscosity values.

Initial perturbation Setup. What are the detailed differences of the medium and large perturbations? (The height of the initial fold core?) Furthermore, how do these perturbations form? (Bedrock topography, variable snowfall, or other mechanisms which can form the initial fold core?) It would be good to point out in the Conclusion (maybe also Abstract) that convection in these models requires an initial perturbation; without it, convection may not initiate (?)

Fold axis direction. The paper primarily presents results in the along-flow direction (e.g., Figs. 4A–D). However, many observed plume-like folds, especially those associated with ice streams and convergent flow regimes, are oriented in the cross-flow direction with fold axes extending along the flow (e.g., Jansen et al., 2024, Nat. Commun.,

https://doi.org/10.1038/s41467-024-45021-8; Franke et al., 2022, Nat. Geosci., https://doi.org/10.1038/s41561-022-01082-2). It would be good to include some statements in the Discussion that, such as, the final plume geometry is associated with the initial 3D perturbation or other three-dimensional mechanisms.

**Specific comments**

Plain Language Summary, Line 8, "where the ice is old (therefore soft) and slow-moving". Why is the old (and slow-moving) ice soft?

Introduction, Paragraph 2, Lines 11-14. "Viscosity gradients" can be deleted because Y. Zhang et al. (2024) claim that viscosity gradients do not play a significant role. Anisotropy alone can produce small-scale folds (fold amplitudes <<100 m) (but I feel you don't need to mention small-scale folds here?). And convergent flow, rheological anisotropy and a rough bed can form large-scale folds (>100 m), but that tall plume-like folds require density gradients …

Figure 1. The captions for panels B and C appear to be mixed up (also in the main text).

Figure 2. Consider marking the location of panel A within Figure 1. Also, please clarify whether vertical exaggeration is applied in panels B-D.

Figure 3. The "red" labels A-E can be labeled as Fig.4A-4E or anything else. It is easy to mix up labels A-E for Fig.3 and Fig.4.

Figure 4E. Convection appears to develop in the cross-flow (y) direction as well. Is there an initial velocity component or perturbation applied in the y-direction?

Discussion, Paragraph 5, Lines 4-8. The overall viscosity values are dynamic along with c-axis rotations, which lead to lower viscosities as a result of directional alignment of the c-axis and anisotropic rheology. This directional softening not only enhances buoyancy effects but also contribute directly to plume growth.

Figure S5. "As for Fig.5 …" I think it is Fig.4?

---

## Author Comment (AC1)

**Response to reviewers for 'Exploring the conditions conducive to convection within the Greenland Ice Sheet' by Law and others**

We would like to thank both reviewers for their time and valuable comments that have helped to improve the framing, discussion, and flow of the manuscript. Also to apologise for the lateness of the reply, I (RL) moved country and had a few other projects to sort out!

The biggest change we have made is to position the results and discussion primairly in terms of effective viscosity variations, rather than the enhancement factor, though we retain the methodolody as a convinient way to shift the effective viscosity profile. We have also made both the model setup, and its limitations clearer, and altered the discussion such that other processes are given additional coverage alongside uncertainties related to our convection model.

We have also created a repository on zenodo. This is still not formally published in case changes are necessary, but it can be found here.

Text inserted during the course of revisions is marked in raw sienna.

Robert Law on behalf of all co-authors October 30, 2025.

**Reviewer 1**

Large plume-like folds are important features observed in the Greenland Ice Sheet, with significant implications for reconstructing past ice dynamics and projecting future ice-sheet mass balance. This paper investigates the formation of these plume-like structures. It is exciting to see the authors apply the well-established geodynamic modeling software ASPECT—commonly used in mantle plume studies—to test the hypothesis that thermal convection can produce such features. The modeling results are robust, as the authors present a comprehensive set of simulations exploring a wide range of enhancement factors, surface velocities, ice thicknesses, snowfall rates, temperature profiles, and model dimensions. Some of the simulated structures closely resemble those observed in radar profiles. The study finds that surface velocity, accumulation rate, and ice rheology exert the strongest controls on plume formation, which can also explain the spatial distribution of plume-like folds across northern and southern Greenland. Overall, this is a well-executed study, particularly valuable for improving our understanding of ice rheology in inland ice-sheet regions, where the ice may initially be frozen on the ground.

Thanks!

**General comments**

The model results clearly demonstrate that thermal convection can generate plume-like folds. I find the numerical results convincing, especially as they align with findings from my own modeling work on buoyancy effects (Y. Zhang et al., 2024). I also conducted tests using lower viscosities that show a more pronounced buoyancy-driven response. I'm excited to see this paper, as it represents a significant step forward in systematically constraining the controlling factors on convection and thoroughly analyzing the formation of plume-like structures.

**Thanks!**

However, I have several concerns when relating the model results to the realistic world: Ice rheology.

1.1. The authors use a Newtonian and isotropic rheology. While the inclusion of an enhancement factor can approximate some aspects of natural ice behavior, this rheology may not be widely accepted. It would be good to include a paragraph at the end of Discussion section to carefully discuss the limitations of this ice rheology to the results. Actually I think a nonlinear and anisotropic rheology is likely to enhance convection, but am not sure how much initial perturbation would be required to trigger convection if the flow law is changed.

We agree that a Newtonian and isotropic rheology is not fully representative of ice-sheet rheology. The problem is that a 2D slice, as used in most of our ASPECT simulations, will not fully replicate the stress-state of an ice sheet. Meanwhile, a full ice-sheet model cannot – under computational limitations – simulate convection. One way to handle this is as in your paper, where a non-linear rheology is still used, but it would not have been computationally for us to test as broad a parameter space as we did under those conditions (100s of individual simulations) and even then it is still not entirely straightforward to emulate a full ice sheet stress state. In certain respects then a linear rheology actually makes the effective viscosity used more transparent (and now displayed more clearly in our Fig. 3 moved to the main text).

This also connects to Point 2.1, and in response to these points (and building on the rationale already provided in the Methods at line 84) we have extended the discussion slightly to include some more caveats.

Starting on line 249: Ice is more accurately represented as a non-linear shear-thinning fluid in most situations, and may also be more non-linear than the linearised approximation of n=3 implemented in this study, with growing evidence for n=4 in some regions

(Bons2018GreenlandMotion, Ranganathan2024ASheets). Our use of Newtonian rheology allows us to test a broad parameter space but may miss non-linear interactions caused by the plumes themselves which both increase and decrease effective strain rates and thereby influence the effective viscosity (Eq. 3). Increasing values of n away from unity may increase the importance of these non-linear stress responses within the plumes; intuitively one may anticipate in a direction that more readily facilitates plume formation, though this will depend upon the appropriate values for  $A_0$  and Z and the resultant effective strain field. We emphasise, however, that rate-weakening in plumes is still expected to be small compared to the main coastward movement of the ice sheet, which exerts a first order control over effective stress (Figs. A2, A4). In any case, we hope that our results closely isolate the effective rheological thresholds for ice-sheet convection, which permits a narrower starting point for future numerical models featuring more complex and computationally costly thermodynamics.

Something that we come back to elsewhere in our response is that it is difficult to say with certainty if anisotropy or highly nonlinear rheology will increase the propensity for convection. As we state above, intuitively one may expect so, but nonlinearity and anisotropy do not in and of themselves say what the effective viscosity will be for a given field as this also depends on the values of A and  $\tau_e$  in Eq. 2. For example, while anisotropy typically leads to bulk softening of ice, it may be appropriate to then begin with a higher bulk volumetric viscosity. And, technically "increasing anisotropy" refers to an increasing contrast in rheological response to a non-symmetric stress state, not a definite softening in one direction. Similarly, a higher n value may require a lower value of A and furthermore influence  $\tau_e$ in ways that are difficult to predict without in-depth modelling. So, while we think our intuition on these matters broadly matches yours, we refrain from speculating too greatly on these points within the manuscript.

1.2. It would also be helpful to explicitly state the range of effective viscosities in the convection models and, if possible, include representative viscosity maps. It is important to know the viscosity values.

We agree, and in response to your comments and those of reviewer 2 we additionally reframe the results mainly in terms of effective viscosity. We still include some discussion of E as E is a single number which will shift the entire effective viscosity profile and is therefore easier to discuss.

Fig. S3 (now Fig. S3) has now been moved to the main text, and expanded to include effective viscosity profiles for E values of 60, 40, 10, and 5 for  $\tau_e$ . In our study the effective viscosity is linked to temperature, not an evolving anisotropy so we think a simple plotted profile is the best way to display this information.

1.3. Initial perturbation Setup. What are the detailed differences of the medium and large perturbations? (The height of the initial fold core?) Furthermore, how do these perturbations form? (Bedrock topography, variable snowfall, or other mechanisms which can form the initial fold core?) It would be good to point out in the Conclusion (maybe also Abstract) that convection in these models requires an initial perturbation; without it, convection may not initiate (?)

The difference between the two perturbations could be seen by comparing Fig. S3 and S4. However, as Fig. S3 has now been moved to the main manuscript, we have additionally plotted the large temperature perturbation alongside the medium perturbation in Fig. A3. The scripts to produce these temperature perturbations are now included in the assembled repository linked at the end of the manuscript.

We mention the requirement for an initial perturbation in the appendices. I (RL) think this was likely moved there just before submission unintentionally but you are correct that it deserves a bit more of a prominent position. We have now made this point (1) at the beginning of the discussion (line 172). We think the processes outlined in your paper offer several mechanisms for the necessary proto-folds. There are also some clear topographic differences that can lead to higher temperatures horizontally adjacent to lower temperatures, and this is in fact one place where plume genesis is seen (Figs. 2B, 2C).

From line 177, this reads: Condition (1) is easily satisfied by bedrock perturbations (e.g. Figs. 2B, C) or basal folding induced by processes such as convergence and anisotropy (Zhang2024FormationSheet).

1.4. Fold axis direction. The paper primarily presents results in the along-flow direction (e.g., Figs. 4A–D). However, many observed plume-like folds, especially those associated with ice streams and convergent flow regimes, are oriented in the cross-flow direction with fold axes extending along the flow (e.g., Jansen et al., 2024, Nat. Commun., https://doi.org/10.1038/s41467-024-45021-8; Franke et al., 2022, Nat. Geosci., https://doi.org/10.1038/s41561-022-01082-2). It would be good to include some statements in the Discussion that, such as, the final plume geometry is associated with the initial 3D perturbation or other three-dimensional mechanisms.

This is a good point, thanks. We have included an additional section in the discussion regarding convergent flow regimes (which also includes some discussion related to reviewer 2's point 2.20 that plumes are found outside the  $1 \text{ m a}^{-1}$  contour)

From line 188 this reads: Relatedly, from Eq. 2 effective viscosity is proportionally related to effective strain as  $\eta \propto \dot{\epsilon}_e^{(1-n)/n}$  or, if n=3, as  $\eta \propto \dot{\epsilon}_e^{-2/3}$ , where  $\dot{\epsilon}_e^2 = \frac{1}{2} \mathrm{tr}(\dot{\epsilon}^2)$  is the effective strain rate. This creates another balance whereby increasing the effective strain (or effective stress through  $\tau_e \propto \dot{\epsilon}_e^{1/n}$ ) reduces the effective viscosity, encouraging convection, but also increases ice-column disturbance, discouraging convection. While our experiments focus on along-flow slices, it is possible that this may assist explanations regarding the presence of tall plumes with off-axis orientations just outside ice stream margins (Frank2022GeometricDynamics, Jansen2024ShearAgo) where effective strain is elevated, and hence effective viscosity reduced, but through-column horizontal shear is not excessive (rather, rotation increases effective strain while not significantly disrupting plumes)..

**Specific comments**

**1.5.** Plain Language Summary, Line 8, "where the ice is old (therefore soft) and slow-moving". Why is the old (and slow-moving) ice soft?

We are removing the plain language summary now that the article is formated for The Cryosphere.

1.6. Introduction, Paragraph 2, Lines 11-14. "Viscosity gradients" can be deleted because Y. Zhang et al. (2024) claim that viscosity gradients do not play a significant role. Anisotropy alone can produce small-scale folds (fold amplitudes ¡¡100 m) (but I feel you don't need to mention small-scale folds here?). And convergent flow, rheological anisotropy and a rough bed can form large-scale folds (¿100 m), but that tall plume-like folds require density gradients . . .

**Done!**

**1.7.** Figure 1. The captions for panels B and C appear to be mixed up (also in the main text).

Fixed, thanks

**1.8.** Figure 2. Consider marking the location of panel A within Figure 1. Also, please clarify whether vertical exaggeration is applied in panels B-D.

Clarified that exageration is applied to panels B-D, also. Thanks for the suggestion on marking this in Fig. 1. We tried this, but found it too noisy, so we have kept this figure as is.

**1.9.** Figure 3. The "red" labels A-E can be labeled as Fig.4A-4E or anything else. It is easy to mix up labels A-E for Fig.3 and Fig.4.

Fixed, thanks!

**1.10.** Figure 4E. Convection appears to develop in the cross-flow (y) direction as well. Is there an initial velocity component or perturbation applied in the y-direction?

We think this may have arisen from a typo in Fig. A3 where we specified that  $v_x$  rather than  $v_y$  was set as zero, this has now been rectified and we have also added dimension labels. We get a distinctive mushroom shape in the y direction (discussed in the text), but we don't see a "roll over" in the y direction, and there is no applied velocity component in the y direction. The small bump visible in Eiii just next to the "3D" text is just a small perturbation in the flow field as the simulation gets going that doesn't influence the results.

**1.11.** Discussion, Paragraph 5, Lines 4-8. The overall viscosity values are dynamic along with c-axis rotations, which lead to lower viscosities as a result of directional alignment of the c-axis and anisotropic rheology. This directional softening not only enhances buoyancy effects but also contribute directly to plume growth.

Thanks, and we agree this is an important point. Please see our response to Point 1.1, but we have essentially re-writen the paragraph discussing anisotropy in response to this Point and Point 1.1.

From line 234 this now reads: In situ rheology is also modulated by anisotropy, which is not included in our simulations. (Zhang2024FormationSheet) suggest an important role for anisotropy in the formation of large plume-like folds (their Fig. 4) as a result of direction-dependent softening due to directional alignment of the c-axis. Plume-forming motion will rotate initially bed-planar fabric such that it also broadly aligns with the dominant shear direction in plume formation. However, anisotropy itself describes a stress-orientation dependent rheology, rather than a specific softening. The role of anisotropy in (Zhang2024FormationSheet) then comes in part from their implementation which allows the viscosity acting along the plane perpendicular to the c-axis maximum (denoted as  $\eta_2$ ) to decrease by a factor of three and fall below the  $1 \times 10^{13}$  Pa lower limit set for the

isotropic run (their Table S2). Such a decrease is sufficient to reach the effective basal viscosity values ( $\sim 3 \times 10^{12}$  Pa s) in our E=40 and E=60 simulations, where local convection becomes increasingly viable. This is not to say that progressive anisotropic softening is not an important process here. In glacier settings, bulk viscosity will generally decrease as ice develops stronger crystallographic anisotropy, though the effect is stress-state dependent (Azuma1994ASheets), meaning such softening may also be a contributing factor for plume locations occurring at distance from ice divides. In our application, as in many others (Cuffey2010TheGlaciers), the enhancement factor operates as a simple parameterisation of anisotropic effects without recourse to a stress-orientation dependent tensorial flow law. Consequently, these results provide a plausible estimate of how progressive anisotropic softening may affect  $\eta_2$ , which likely exerts the strongest rheological control on convection onset.

**1.12.** Figure S5. "As for Fig.5 ..." I think it is Fig.4?**

Thanks for catching that – corrected.

**Reviewer 2**

In this paper, Law and coauthors use geodynamic modeling techniques to explore the possibility that local thermal convection may occur in the Greenland Ice Sheet. Thermal convection has long been hypothesized for continental ice sheets, mostly by Terry Hughes, yet it has never seen observational confirmation or enjoyed widespread acceptance in the glaciological community. However, over the past decade or so a growing literature has developed around the observation of large plume-like folds in northern Greenland. There is no consensus on the formation mechanism of these plumes, with proposed causal mechanisms including basal freeze-on, traveling slippery patches, rheological contrasts in the ice column, and cross-flow convergence. This paper adds to the literature on these enigmatic plumes by proposing that they may have formed through buoyant thermal convection, thus posthumously vindicating Terry Hughes' long and lonely crusade to gain support among glaciologists for the existence of thermal convection in ice sheets. The bulk of this paper is spent describing and analyzing the results of applying a geodynamic model to Greenland-like conditions. The authors' model results indicate that thermal convection may in fact be possible under the conditions that prevail in north Greenland. They find that the development of convection is inhibited by higher surface accumulation rates and faster flow speeds, explaining why plumes are not observed in south Greenland. Their model requires ice viscosity significantly lower than standard rheological assumptions, which, if true, would have important implications for the modeling of ice sheet dynamics and for the results of basal traction inversions.

**Major Comments**

This paper is clearly appropriate for The Cryosphere. It represents both an important addition to the literature around a relatively new observational mystery (the large englacial folds) and a dramatic coda to a very old theoretical debate (thermal convection). The paper is well written and argued. Some of the figures need work in order to better display the authors' results. In particular, they seem to have crammed their entire parameter space exploration into Figure 3, which is quite busy and difficult to understand. However, they only have four figures total in the main text, so they have plenty of room to split this information into an additional figure to aid comprehension. Their model has some issues, which I discuss next, that potentially undercut their conclusion that rheology must be softer than commonly assumed. However, their model also has some strengths relative to typical ice sheet models, and their overall conclusion that thermal convection can explain the observed plumes is supported by the model results presented. My concerns about the simplifications made in their model mainly fall under the heading of, "all models are wrong, but some models are useful", and my concerns can be addressed through changes in the discussion or conclusions text, rather than a redo of the modeling itself. My recommendation is to publish with minor revisions.

**Thanks!**

I now go on to discuss my main concerns with their model setup.

**2.1.** For rheology, they used a linear Newtonian rheology for ice, derived from the non-Newtonian rheology by assuming a constant effective stress ( $\tau_e = 50 \text{ kPa}$ ) throughout the domain. They include the temperature dependence of rheology, but not the stress (or equivalently, strain rate) dependence. They justify this omission by claiming that the strain rates associated with convection are much lower than the background strain rates associated with horizontal ice flow, but the background strain rates should be highly concentrated near the bed, while they have used a constant 50 kPa value for  $\tau_e$  rather than a vertically variable one. It is not necessarily clear to me that the strain rates associated with convection will be lower than the background strain rates in the midcolumn, and the authors have not shown this comparison to justify their assumption. While I do not believe that the use of a linearized rheology undercuts their conclusion that thermal convection is possible for the conditions that prevail in north Greenland, it does undercut their conclusion that the enhancement factor must be much larger than typically used in ice flow models. The authors have concluded from their model results that E must be an order of magnitude larger than the typical value, but their results could just as easily be interpreted to mean that they used a value of  $\tau_e$  that was too small. Additionally, it is not necessarily clear that the particular thresholds for convection that they found when varying E would hold up if they had used either a vertically variable  $\tau_e$  or a fully nonlinear rheology.

Please see Point 1.1 for a brief discussion of why a linear approximation for rheology was necessary in this case. We agree with the issues regarding  $\tau_e$ . As covered in our reply to Point 1.1 we have retained the  $E, \tau_e$  combination (and with  $\tau_e = 50$  kPa for consistency) for some of the reporting as it provides a simple way to shift the entire viscosity profile (and is linked through Eq. 3). But we have shifted the emphasis of our reporting towards the range in effective viscosities in the required profiles throughout the manuscript.

Thanks also for the comment regarding our use of a constant effective stress. We agree that this at first glance does not seem super intuitive so we gave this some considerable extra thought following your comments. Basically, effective strain will of course increase towards the bed, but so does temperature. This means viscosity decreases towards the bed such that we come to the result that  $\tau_e$  actually doesn't exhibit a great deal of variation with depth.

We include this in the appendix discussing ISSM (and referenced in the methods at line 85) at line 309:  $\tau_e$  remains surprisingly uniform throughout the ice column, justifying our use of a constant value (and, deciding on the form of a variable  $\tau_e$  profile with depth would present its own issues, hence we opt for the simplest approach for transparency). This can be understood in part through the relationship

of  $\dot{\epsilon}_e$  and A with depth. We can set  $\dot{\epsilon}_e = A\tau_e^n$  or  $\tau_e = (\dot{\epsilon}_e/A)^{1/n}$  and then  $\ln(\tau_e) = 1/n(\ln(\dot{\epsilon}_e) - \ln(2A))$  which gives:

$$\frac{d}{dz}\ln(\tau_e) = 1/n(\frac{d}{dz}\ln(\dot{\epsilon}_e) - \frac{d}{dz}\ln(A)). \tag{1}$$

This means that the variation in  $\tau_e$  with depth arises from the difference between the log of  $\dot{\epsilon}_e$  and A, but as both are increasing with depth in our setup the variation is not that large. For example, both  $\dot{\epsilon}_e$  and A may be expected to vary by one order of magnitude between -15 and -1 deg C and half and full depth, respectively.

Therefore, while some vertical variation in  $\tau_e$  is to be expected the magnitude is not that high and we feel that the constant approximation reflects the ISSM modelling and has a rational explanation.

**2.2.** For the boundary conditions on the upper surface, they used Dirichlet conditions to impose both horizontal flow and surface accumulation, rather than having a stress-free free surface like the real ice sheet. This could have been more problematic but did not end up being a huge issue because they included the temperature dependence of rheology, so shear was concentrated near the bed anyway.

Yes, adapting convection code that works very well for mantle systems but was not built with glaciers in mind was a challenge, but one we hope we have tackled appropriately!

2.3. A bigger issue is the lower surface: they did not state what boundary conditions they used for temperature, but I was able to infer from other parts of their model setup and results that they used a Dirichlet condition with basal temperature set to -2°C, which is a reasonable approximation of the pressure melting point under 2 km of ice. It is likely that they did this because the classic convection problem in fluid mechanics or geodynamics involves a fixed  $\Delta T$  across a specific layer thickness. However, in ice sheets, the basal boundary condition is typically Neumann (gradient determined by geothermal heat flow) up until the point that the basal temperature reaches the melting point, when it switches to Dirichlet. Thus, the authors' use of a Dirichlet condition for temperature at the ice base implies that the ice base is wet, but this contradicts statements in the paper that the authors believe that the bed is frozen where the plumes are observed, and also contradicts the authors' use of a no-slip condition at the ice base.

Thanks for pointing this out, which comes up in other parts of your review, too. Essentially, this was a small oversight on our part. Below  $\sim 3$  km depth -2 °C will be below the freezing point, and our base simulation thickness is 2.5 km. This means that for the majority of our

simulations the base temperature is below the pressure melting point, only in simulations with variable thickness where the depth is greater than  $\sim 3$  km is this condition invalidated. As sustained or amplifying convection is already well underway by this thickness (Fig. 4G,H,I in the updated manuscript) we are not overly concerned about this small misrepresentation.

At line 118 we write: Above  $\sim 3,000$  m, the basal temperature is then technically above the pressure-melting-point, even though a non-slip condition is imposed at the base at all times. Simulations where H > 3,000 m comprise a small proportion of our overall ensemble and we do not change the basal temperature for this subset.

We also make the boundary conditions more explicit, writing at line 116: (Dirichlet condition, therefore indirectly neglecting geothermal heat flux). And at line 128: and Figs. 3 and A3 illustrate boundary conditions.

**2.4.** The presence of sliding at the ice base does not necessarily undercut the authors' conclusion that thermal convection is possible; if anything, since they found that vertical shear suppresses the development of convection, allowing basal slip may actually broaden the parameter range over which convection is possible. However, the presence of sliding may undercut their rheological conclusions, where they argue that sliding is less extensive than existing inversions have found. Also, if the basal temperature is tied to the melting point, then the temporal and spatial variations in conductive heat flow that they found (Figures 4 and S7) should be associated with changes in the melting or freezing rate, which should impact the availability of basal water and the basal traction. Those changes in the basal boundary condition should, in turn, influence the ice flow field. Thus, we would expect thermal convection to interact with both the basal freeze-on and traveling slippery patches mechanisms. In order to truly model convection alone, the authors would need to use Neumann conditions for temperature at the ice base, and choose a parameter range where they were sure that the base would never warm to the melting point. A model of thermal convection in ice sheets with a warm bed but no sliding and no variations in basal melting or freezing is necessarily omitting some pretty important feedbacks.

See above (Point 2.3) for discussion regarding the -2°C value. We definitely do not include sliding – we did think about it, but then a sliding relationship must be introduced and this makes it (1) harder to control the strain profile of the ice which is the important control on plume dynamics and (2) would be challenging to implement in ASPECT. This does not mean we say no sliding can occur within ice where plumes are occuring. Sliding in effect would be a translation of

the entire model domain (and some associated component of stretching, perhaps). As we state on line 64:

Therefore, while increasing  $v_{x,s}$  in the model can simulate plume behaviour as *actual* surface velocity increases (Fig. 1C), the comparison is not one-to-one – our modelled plume behaviour represents the upper, not lower, limit of disruption that can occur under a given  $v_{x,s}$ .

As also addressed in Point 2.3, we have clarified the boundary conditions more clearly, both in the text and by moving Fig. S3 to be Fig. 3.

2.5. To be clear, I do not think that there is anything wrong with writing a paper focused on convection alone, especially considering convection's contentious history in glaciology, and considering the fact that the original papers on basal freeze-on and traveling slippery patches (which I was involved in as either first author or coauthor) completely ignored thermal buoyancy. However, when discussing results and drawing conclusions, it is very important to make note of what processes were omitted from the model, and to think about how those processes might affect the results. In reality all processes are coupled together, and we cannot think of convection as being truly independent of basal freeze-on or traveling slippery patches (or, for that matter, from anisotropy and cross-flow convergence). In the real world, the basal boundary condition is not fixed and immobile; rather, the variations in the englacial temperature field seen in the authors' model will produce spatial and temporal changes in the conductive heat flux at the ice base, which produces changes in the melting or freezing rate, changing the availability of basal water and basal traction, which finally feeds back on the original englacial flow field produced by convection in the first place. Even if the background temperature field is initially cold-based, the uplift underneath a rising buoyant plume is likely to have a lower conductive gradient than its surroundings, warming the bed locally and potentially producing a local patch of basal melt. Do these feedbacks act in a way that amplifies convection, or suppresses it? Will they broaden the parameter range over which convection is possible, narrow it, or merely shift it?

As I said previously, I think that my criticisms of the authors' model setup mostly fall under the old saying, "all models are wrong, but some models are useful". The authors used a model that is well-designed for simulating convection and poorly designed for other things. In most geodynamics problems there is no need to simulate a dynamic basal boundary that switches modes in response to changing conditions within the model domain; thus, the authors' model can't include subglacial hydrology. This doesn't mean that the authors' results have no value. On the contrary, they have showed, at long last, that thermal convection is possible in ice sheets, and they have connected this model result with an enigmatic set of observations that could plausibly be caused by convection. This is an important result and worthy of publication. My main concern is that the

discussion and conclusions sections need to be more forthright about the limitations of using a geodynamics model for glaciological problems, and these sections should also discuss potential feedbacks between thermal convection and other mechanisms that might contribute to the folds, especially mechanisms that are sensitive to the thermal state of the ice base.

Thanks for the comment and yes we agree that "all models are wrong, but some models are useful" – hopefully this one is well placed on the useful side! We are also, of course, happy to discuss these points. We have kept **the lines:**

Travelling slippery spots may also not be compatible with the 3D geometry of observed plumes (Figs. 2B, D) or with ice motion over a rough bed. Additionally, neither mechanism accounts for an apparent absence of H > 1/3 plumes in south Greenland.

The main reason these points are included is out of genuine curiosity – addressing these questions is clearly beyond the scope of the present manuscript, but we do believe fully understanding plumes is a good way to understand ice thermodynamics more generally, and we think that these questions are very legitimate avenues for further research.

We follow your suggestion in including some further discussion about model limitations at line 214:

However, the basal thermal state may have been different  $\sim 10$  kyr ago and we do not rule out these two processes contributing to the onset of an initial perturbation or playing a role in their continued development. Our imposition of a no-slip, constant-temperature basal boundary also means that possible feedbacks between convective heat dispersal and basal sliding are not recognised. These may complicate plume geometry in a similar manner to that explored in (Wolovick2014TravelingSheets).

We hope these changes adequately address these important points.

2.6. Finally, I think that the authors' conclusion about the likely value of the enhancement factor E should be given more caveats. Given the simplifications that they made to ice rheology in their model setup, and given that they omitted feedbacks that could potentially change the parameter range over which convection is possible, I am not convinced that we can necessarily use the observed plumes to infer that ice is softer than commonly believed. The authors are free to keep arguing that, of course, but I think

some caveats are necessary for that particular conclusion.

In common with our responses to other instances where the enhancement factor is mentioned (Points 1.2, 2.1), we have placed much more emphasis on the effective viscosities required and use this as the 'head-line' for our conclusions, though retain some usage of E as a helpful descriptor of the entire profile.

**Minor comments**

**2.7.** Note that my copy of the manuscript does not have line numbers. I will try to place my comments by giving the section of the paper and a quote.

Line numbers added and Cryosphere template now being followed.

**Introduction:**

2.8. "At first glance, an entirely separate problem is the nature of the formation of large (¿1/3 the local ice thickness) englacial plumes found by tracing reflections of equal age in radargrams (i.e., isochrones; Figs. 1A, 2, S1, CReSIS (2013))"

Add a reference to Bell et al. (2014) here. You have a reference to Bell elsewhere in the paper, but Bell et al. (2014) was also the first to describe the large plume-like reflectors in northern Greenland and they should be cited here, even if you don't agree with their interpretation of the reflectors as originating with freeze-on.

In response to Points 1.2, 2.1, 2.6 we have changed the introductory paragraphs somewhat, but we have fixed this omission in the first paragraph

2.9. "Such plumes have previously been hypothesized to result from basal freeze on (Leysinger Vieli et al., 2018), or traveling basal slippery spots (Wolovick et al., 2014), which both require an at least temporarily thawed bed."

The end of this sentence is awkwardly phrased. It also misses the issue of spatial variability, which is important to both mechanisms in addition to temporal variability. Maybe try, "...which both require that the bed be at least locally or temporarily thawed".

**Fixed!**

2.10. "Both authors approach convection analytically only, by estimating a Rayleigh number Ra, the dimensionless ratio of heat transfer via upwards mass transport (i.e. convection) vs. thermal conduction (Rayleigh, 1916)...The formulation of thermal diffusion in Ra does not capture dynamical effects important in ice sheet flow, such as horizontal shearing; and the critical Ra is itself tied to the particular boundary conditions and the

initial perturbation geometry"

Additionally, this analytic approach misses the fact that heat is already being advected by mass transport within the ice sheet- vertical thinning and subsidence associated with surface accumulation is pushing colder ice down. Any upwards convection needs to fight against this background state of subsidence, as you found in your model results later.

Yes, agreed. At line 43 we have addded: or downwards motion from snowfall.

**Materials and Methods:**

**2.11.** "Surface mass balance is set to zero, i.e., no snowfall or surface melting (vz,s = 0 at the surface boundary condition . . . surface shearing velocity vx,s is uniform across the domain's top surface"

Does this mean that you apply Dirichlet conditions to velocity at the upper surface of the domain, rather than stress-free (Neumann) conditions? Does this also mean that your upper surface is a rigid lid, rather than a free surface?

Yes, your interpretation is correct, and we have made this clearer in the text by adding at line 60:

This Dirichlet velocity condition on a fixed surface differs from a 'standard' ice sheet model, where surface velocity is an emergent result of ice geometry and flow parameters, but is suitable for our purposes as we treat surface velocity as an independent variable in simulations. Regardless, the net effect on background (i.e., not convection controlled) stress and strain fields is similar.

The correct boundary conditions are Neumann, and the correct way to induce horizontal flow is to have a slope in the free surface that drives ice flow downstream, and the consistent way to induce vertical flow is to have the gradient of this induced horizontal flow have an along-flow gradient that balances the accumulation rate. I suspect that you have done things this way because your geodynamic model was not built to study ice dynamics, and so the steady state that I just described is beyond its ability to compute.

Correct. The approach you suggest is just about workable, but comes with its own compromises and problems. Importantly, as we wish to test velocity, height, and snowfall as independent variables, prescribing these as Dirichlet conditions is the most straightforward option. As we cover in the discussion, we hope that the results we present here may form the basis for future work that can use our parameter exploration

as a starting point.

I don't think that this necessarily invalidates your results, mostly because Figure S5 shows that horizontal velocity is mostly uniform in the upper column, with shear mostly concentrated in the lower column. If your model had produced too much shear in the upper column, I would suggest throwing the whole thing out. However, you need to explicitly state that you use Dirichlet conditions rather than Neumann conditions at the upper surface, and you need to have some text justifying this choice and explaining what impact (if any) it has on your results.

As covered above and in other response we have made this cleare in the methods and discussion.

In addition, you do not state what boundary conditions you use for temperature on the lower surface, but because heat flow across the lower boundary varies in time (Figure S7), I infer that you also used a Dirichlet condition for temperature, with a value of -2°C (roughly accurate for the melting point underneath 2 km of ice). I discussed my issues with this at greater length in the Major Comments section above; for here, my main comment is that the basal boundary condition needs to be stated explicitly.

Please see our responses to other Points.

**2.12.** "A Newtonian rheology is appropriate here as strain rates due to convection are small compared to those from background ice flow (Fig. S5)"

Figure S5 does not show strain rates, it shows velocities. I would like to see a figure for strain rates to support this argument. In particular, I am curious about whether the strain rates due to convection are smaller than the background strain rates in the mid-column, not just near the bed. This is potentially an important caveat to your results and an important limitation of trying to use a Newtonian rheology to model a non-Newtonian material.

See also Point 2.31. We have added new panels to Fig. A4 (previously Fig. S5) to show strain rate. In the case of amplifying convection (Panel Cvi) convection-driven strain rates are clearly somewhat above (or below) the background mid column strain rate. However, this is not the case for the 'sustained' bracket (Panels Avi, Bvi). We have added to the caption of Fig. A4:

For Panel Avi, the effective strain rate in the mid column of  $1\times10^{-12}$  may increase by 50% or decrease by 20% due to convection. For panel Cvi the effective strain in the mid column of  $3.5\times10^{-12}$  may increase by 100% or decrease by 60% due to convection.

Of course, any deformation within the column will result in a change in the strain rate tensor and therefore the effective stress. Our approximation is not perfect, but we think it's appropriate in this setting, and a very big leap forwards from annalytical methods. We quote the full altered paragraph below **starting at line 249**:

Ice is more accurately represented as a non-linear shear-thinning fluid in most situations, and may also be more non-linear than the linearised approximation of n=3 implemented in this study, with growing evidence for n=4 in some regions (Bons2018GreenlandMotion, Ranganathan 2024 A Sheets). Our use of Newtonian rheology allows us to test a broad parameter space but may miss non-linear interactions caused by the plumes themselves which both increase and decrease effective strain rates and thereby influence the effective viscosity (Eq. 7). Increasing values of n away from unity may increase the importance of these non-linear stress responses within the plumes; intuitively one may anticipate in a direction that more readily facilitates plume formation, though this will depend upon the appropriate values for  $A_0$ and Z and the resultant effective strain field. We emphasise, however, that rate-weakening in plumes is still expected to be small compared to the main coastward movement of the ice sheet, which exerts a first order control over effective stress (Figs. ??D, 3, A4). In any case, we hope that our results closely isolate the effective rheological thresholds for ice-sheet convection, which permits a narrower starting point for future numerical models featuring more complex and computationally costly thermodynamics.

**2.13.** "Prescribing  $\tau_e$  is also necessary as a full ice-sheet stress state can not be accurately replicated in a simplified along-flow slice"

Especially when you don't have a true free surface and instead produce horizontal flow by imposing a Dirichlet condition at the top of your domain.

Yes we agree. Please see our responses to other Points for how we have updated the description/discussion surrounding our ASPECT model.

**2.14.** "We set Fint = 0, thereby ignoring adiabatic heating and neglecting strain heating, to prevent simulations with greater vx,s and hence greater strain heating from evolving a different rheology along flow."

This is a decent first approximation for the slow-flowing areas of the ice sheet, but it is potentially an important limitation. At a velocity of 10 m/yr (appropriate for many of the observed plumes) and a driving stress of 100 kPa (a good ballpark number for ice sheet stresses in general), the integrated shear heating is 32 mW/m2, or about half of

the geothermal heat flow. This could potentially play an important role in warming and softening the basal ice, and also could contribute to convection by providing additional thermal energy in the lower ice column that needs to escape to the surface. At a velocity of 1 m/yr that would be reduced by an order of magnitude, but as I said, some of the plumes are observed where velocities are about 10 m/yr or even higher.

We agree that this is a reasonble first approximation. Please also see our responses to Point 1.4 and 2.22 regarding the 1 m a-1 values. In response we have added **at line 104**:

though we note that such heating will soften ice and may further facilitate convection.

**2.15.** "We apply a transformation, T2 = ((T1+Ta)/Tb)(Tb + Ta) where T1 is the original temperature profile, Tb is the basal temperature and Ta is an adjustment term used to raise the basal temperature to -20 C. The temperature profiles are stretched and compressed when adapted to the range of ice thicknesses."

Two issues: 1) does this transformation leave the surface temperature unchanged, or are you also changing the surface temperature?

Yes, this will modify the surface temperature unless the surface temperature is  $0^{\circ}$ C. We have not added this into the manuscript as this can be checked by hand by an interested reader. Justification for this adjustment is provided in the sentences following the equation. We are after a reasonably representative temperature profile, somewhat downstream of a core location, not at the core location itself.

2) More importantly, this temperature profile is not going to be in steady state with the enforced accumulation rate and ice thickness in your model. Your Figure S7 shows a pretty dramatic initial adjustment of the domain-average basal conductive heat flow. That could be because your initial temperature is not in steady state with the downwards advection that is actually in your model. Why did you use this approach instead of computing a steady state vertical temperature field for your given accumulation rate and ice thickness?

We wanted to maintain uniformity between simulations and computing a steady state vertical temperature field would have required variation in the heat flow for each snow fall value. As in other isntances, this is a compromise which we believe to be appropriate for our uses even though we accept that there are different ways of implementing things.

It is true that you don't necessarily know the shape function for vertical velocity until you actually run your model, but you can still get much closer to a steady state by applying

a simple approximation (like a Nye model) rather than simply scaling the DYE-3 or NEEM profiles, which come from particular locations where the accumulation rate and ice thickness do not necessarily match the values used in your particular experiments.

This is a valid point, but as with our responses above we just want reasonable representative values that can be applied across all simulations. No temperature profile will perferctly reflect all conditions and locations (and most locations are completely unconstrained by observations), so we think our approach is well-justified in this instance. As we state at state at line 257, we hope our findings (1) demonstrate that convection is possible, and (2), provide a useful starting point for studies probing these processes further when including more processes. Now that this work is (almost) completed, it is easy to look back and see that the hypothesis makes sense, but when we first embarked on this project it seemed like a slightly mad idea and our main priority was to see if convection was at all possible under realistic conditions and then if so, to document what those were.

**Figures:**

**2.16.** Figure 1.

Notes: 1) the descriptions for subplots b and c are swapped.

Fixed

2) It would probably be good to reproduce the observed plume locations in every plot, not just a.

We tried this but did not implement it as it makes the plots too busy.

3) A shape factor of 0.8 is appropriate for n=3 rheology and constant temperature. With a more realistic temperature structure, shear should be more concentrated near the bed and the shape factor should be closer to 1.

This is largely for illustrative purposes. With only six panels we have chosen not to produce an additional panel with a shape factor of 1.

4) Why evaluate effective stress at 5/14 depth? Where does this number come from?

This is largely an arbitrary choice that results from displaying a 3D field in 2D, but is logical as it represents just over the 1/3 height we are interested in.

5) Subplot e might be better on a log scale, since we are mostly interested in areas where the accumulation rate is quite low. Either that or just reduce the maximum of the color

scale.

We played around with a few values here. We believe the existing version works well, particularly with the contours included. We have however added to the caption: The two grey lines represent contours of 0.15 and 0.25 m.w.e  $a^{-1}$ .

**2.17.** Figure 2.**

These echograms are way too dark, at least on my screen. It is very hard to see anything. You should adjust the color limits to improve visibility. The oblique view is also pretty hard to make sense of. Why do you display the entire ice sheet, instead of just zooming in on the region of interest?

We have adjusted the colour limits such that the contrast is now clearer. The entire ice sheet is displayed to make the location of the transects more apparent (see also Point 1.8).

**2.18.** Figure 3.**

There is a lot going on in this figure and it is hard to interpret. I would recommend splitting this figure into two figures. For one thing, the use of lines when you have timeseries data, but the lines don't actually represent progress over time, is very confusing. I would recommend that one figure be timeseries (ie, the x-axis should be time, with color representing some other parameter) so that the reader gets a sense of how the model evolves over time. Then another figure could focus on the parameter space exploration by showing 2D contour plots at various cross-sections through your 4D parameter space (the 4 dimensions being enhancement factor, surface speed, height, and snowfall). The second figure should not have any time dependence in it, you should just choose a single metric to quantify the strength of convection (based on the final paragraph of the methods section this would either be the max vz at 20 ka or the change in max vz between 4 ka and 20 ka). Thus, the first figure gives the reader a sense of model evolution for a handful of representative parameter values, while the second figure shows the reader how the strength of convection varies as a systematic function of parameter space. But as it is now, Figure 3 tries to show both an exploration of parameter space and evolution through time, and the result is that there is simply too much going on in one figure.

We do take this comment seriously, and think good figures are central to the presentation of a paper. This figure displays a lot of data central to the story of this paper and went through many iterations and we believe that, while it may take a moment to become acquianted with, it does well represent the behaviours occurring through all of our simulations. We prefer one large figure containing all of the information required to interprate the results, rather than several spread over

different pages which require their own separate interpretation times. We have also updated the figure caption to add:

We display two columns for the NEEM temperature profile – beyond the first row these show a difference in E between 40 and 60, while the first row shows the difference between the large (left) and medium (right) temperature perturbations.

**and:**

An effective stress,  $\tau_e$ , or 50 kPa is used in calculating all effective viscosity profiles. See the text for our definitions of Suppressed, Sustained, and Amplifying convection, printed in bold for easy visibility.

**2.19.** Figure 4.**

"a vertical enhancement factor of 2" I think you mean vertical exaggeration? Vertical enhancement factor invited confusion with the rheological enhancement factor E. It might also be a good idea to adjust the color scale for the stratigraphy figures so that the layers have more contrast.

Updated to exaggeration. We think layer stratigraphy is sufficiently visible and was altered carefully at the time of figure creation. Basically, panel Civ is diffuse as the degree of movement results in heightened numerical diffusion of the tracers. Increasing the range used results in a greater proportion of the panel being "maxed out" in black or white.

**Discussion**:**

**2.20.** "(2) Total horizontal shear through the column must be less than around 1 m yr -1"

Many of the plumes (especially upstream of Petermann Glacier) are in ice flowing faster than this.

This relates to our point that the surface velocity in our model is not a 1-1 match with surface velocity in reality (it reflects the upper bound of the disturbance). We have made this clearer at Line 65, and included at line 174:

through the column convection is occuring in

as an additional constraint for condition (3).

**2.21.** "(4) the enhancement factor must exceed around 45-75."

See my major comments about the need for caveats around your rheological conclusions.

```
We have addressed the major comment in our response to Point 2.1. We have rephrased this threshold in terms of effective viscosity so that it now reads at line 176: the effective viscosity profile range should fall within \sim 2\times 10^{12}-3\times 10^{14} (equivalent to an enhancement factor of \sim 45-75 if \tau_e = 50 kPa )
```

**2.22.** "condition (2) is likely satisfied by low surface velocities throughout northern central Greenland"

I don't know about that. The region with velocity less than 1 m/yr is actually quite small. The region below the 10 m/yr contour is bigger, but still doesn't include many of the observed plumes.

Later you suggest that the plumes may have formed further inland before being advected to their present locations, but it is worth emphasizing that the region within the 1 m/yr contour in Figure 1 is actually tiny, and it is associated with escape times of many tens of thousands of years. You also suggest that the plumes may have formed before the Holocene, when accumulation rates were lower, ice was thicker, and (presumably) flow was slower. However, any plume that is being passively advected by ice flow will also be shrunk substantially by vertical thinning: the characteristic vertical strain rate associated with surface accumulation is a/D, where a is surface accumulation and D is ice thickness, and for values of 10 cm/a and 3 km (being generous to plume survival with both parameters!), that is a strain rate of about 3x10-5 a-1, or a characteristic thinning timescale of 30 ka. Thus we would expect a plume that initially formed near the ice divide where flow was around 1 m/yr and then took several tens of thousands of years to be advected to its observed position would have been reduced from its original size by a factor of 1/e. Since the observed plumes are already a third to a half of ice thickness, that is clearly impossible. Less generous assumptions about accumulation rate and ice thickness can easily lead to a thinning timescale shorter than the Holocene.

Please see our discussion regarding the 1 m a-1 value elswhere in our responses (Points 1.4, 2.20). We note in the methods at line 65 that:

our modelled plume behaviour represents the upper, not lower, limit of disruption that can occur under a given  $v_{x,s}$

We have also substantially expanded our discussion on these points. Please see the added text in Point 1.4 in addition to the following at line 185:

This may occur due to the continued disruption of plumes as the velocity field evolves after they have attained their maximum amplitude in a thicker, colder, and larger palaeo-GrIS (Lecavalier2014AExtent) – although this places plumes in a delicate balance between transport downstream and thinning to below the 1/3H observed threshold.

We hope these adjustment address these important concerns.

In addition, arguing that the plumes originally formed where ice flow is less than 1m/yr begs the question, "why don't we observe any plumes within the 1 m/yr contour now?". I suppose it is possible that the plumes formed during the LGM and then plume formation stopped during the Holocene, however, the Holocene stratigraphy above the plumes is deflected upward, which implies that they have been active during the Holocene. In my opinion, it is much more likely that the plumes formed close to their present-day locations (and that some of them are still forming!), where velocities are more likely to be around 10 m/yr rather than 1 m/yr. But in any event, you need to be explicit here that you are making a prediction: your argument implies that the plumes are currently being passively advected by ice flow, not actively forming. This prediction can be checked through the use of pRES data to measure vertical velocities within the ice column.

I do not think that all of this is necessarily fatal to convection as a formation mechanism, since (as I discussed in my major comments above) the presence of basal slip may reduce the vertical shear that tends to suppress convection, and also coupling between englacial convection and subglacial hydrology may modify the parameter range over which plumes form. But in any event, I think that the discussion needs to spend more time dealing with a few facts: 1) all of the plumes are observed where ice flow is faster than 1 m/yr, in some cases an order of magnitude or more faster, and none are observed where flow is slower than 1 m/yr; 2) escape times from the 1 m/yr contour are substantially larger than plume formation times; 3) passively advected plumes should shrink over time in response to surface accumulation and vertical thinning; and 4) Holocene layers above the plumes are deflected upwards. How do you square these facts with your model result indicating that convection requires velocities less than 1 m/yr, and your argument that the plumes initially formed during the LGM and are only being passively advected during the Holocene? Or, if you think that the plumes are being actively formed during the Holocene, then why don't we observe any near the divide, and why do we only observe them where ice is flowing faster than your model says should be possible?

Thanks, and we agree that the discussion can be expanded regarding these points. Please see our earlier response to this Point, and to Point 1.4 and the changes to the discussion detailed there. We agree that repeat radar surveys can help determine if these plumes are active (and there may be hints of that...). We put this into the text at line 196:

Modelled upwards velocity rates may push through pRES measurement and location uncertainty making analysis of repeat radar surveys a feasible way to test if these plumes are actively expanding.

We thought about putting numbers on what this pRES uncertainty is, but believe that's actually a complicated question that is better left to a future study.

Also, to mention briefly, we agree that slip can extend the range of where these plumes are possible, but so too can enhanced deformation towards the bed. That is, if the deformation is acting in a localized zone *beneath* the plumes, then this functions in the same way as sliding, just over a non-planar (but still potentially thin) region.

**2.23.** "Traveling slippery spots (Wolovick et al., 2014) develop clearly in an controlled setup, but also require thawed bed areas in the same region..."

True, but I don't see why that's a problem. There's plenty of uncertainty in the basal temperature and its not unreasonable to postulate at least local areas of thawed bed in the interior of Greenland. Plus, your model setup has the basal temperature tied to the pressure melting point too.

Please see our response to Point 2.5 where we hope we have addressed this point! As stated elsewhere, discussion of previous papers is largely to motivate further research into this interesting topic.

**2.24.** "...and further do not appear to align with the observation of a highly deformed basal layer beneath the plumes (Fig. 2C), which may be more consistent with high rates of basal ice deformation than basal sliding (Y. Zhang et al., 2024)."

The traveling slippery spots mechanism does not require 100% basal slip in the slippery patches, just a decent contrast in slip percentage between the slippery and the sticky patches. Wolovick and Creyts (2016) goes into more detail about this, and also shows that the overturned parts of the folds are more likely to be found over the sticky parts of the traveling stick-slip trains anyway (DOI:10.1002/2015JF003698).

Thanks for these comments which are keeping us on our toes! Essentially, yes we agree with the second paragraph above, but our point here is that travelling slippery spots do require *some* basal slip, even if it is quite a long way away from 100% of surface velocity. The feature we point out in Fig. 2C is the highly disrupted layer of non-negligible

thickness between the base strictly, and the clear radar interface that demarcates the plumes. While some plumes do clearly enter into this layer (Fig. 2C), some don't. On the other hand, in Wolovick and Creyts (2016) all layer disturbances go straight down to the flat bed (their/your Fig. 6). This discussion is maybe a bit too detailed for the paper, and our main focus is on convection processes, so we don't include this (though happy to talk more about this and other processes outside of the review process at a later date) but we have **added at line 210**:

**to facilitate at least a degree of slip**

to make it clearer that we are referring to any slip, rather than just a high degree of slip with respect to surface velocity.

More speculatively, perhaps local ice hardening could provide a mechanism for travelling slippery patches in the absence of strict sliding? Not sure exactly how that would play out though!

**2.25.** "we cannot rule out these two processes contributing to the onset of an initial perturbation."

That's a good start, but I want to see greater discussion of the ways that convection can interact with other proposed mechanisms. Not only traveling slippery spots and basal freeze-on, but the explanations based on rheological contrasts (DOI:10.1038/nature11789) and cross-flow convergence (DOI:10.1038/ncomms11427) too. I realize that you do talk about anisotropy later in the discussion, but mostly in the context of whether a softer rheology is reasonable, not in terms of formation mechanisms for the plumes.

Thanks for pushing on this. In part this Point is addressed in our response to Point 1.4. In addition, we add at line 216:

Our imposition of a no-slip, constant-temperature basal boundary also means that possible feedbacks between convective heat dispersal and basal sliding are not recognised. These may complicate plume geometry in a similar manner to that explored in (Wolovick2014TravelingSheets).

We give some further space for the Bons and Neem papers you have linked **at line 218**: Rheological contrasts (Dahl-Jensen2013EemianCore) and convergence (Bons2016ConvergingSheet), as covered in Zhang2024FormationSheet may also interact with convection.

**2.26.** "We emphasize, however, that rate-weakening in plumes is still anticipated to be small compared to the main coastward movement of the ice sheet which exerts a first order control over effective stress (Figs. 1D, S1, S3)."

First of all, I don't really see how Figs S1 and S3 address this question. Second of all, as I discussed in my major comments, I buy the argument that the effective stress is dominated by the background flow of the ice near the bed, but I am not convinced that this is the case in the mid-column. I would like to see a supplemental figure showing strain rates in the model, in addition to Fig S5 which shows the components of velocity. The plumes are a relatively short-wavelength feature with vertical flow rates quite a bit larger than the vertical flow rates due to accumulation rate, so I strongly suspect that they are the dominant component of the strain rate tensor in the mid-column.

Thanks. Yes we agree that Fig. S1 (now A1) is not useful in arguing this point and we have removed reference to it (likely a typo). S3 (now A2) is included to emphasize the role of effective stress within the ice sheet. Please see our responses to your major points for further discussion on the effective stress.

**Appendices:**

**2.27.** Appendix A1:**

"If we extend the basal viscosity (calculated at  $5\times104$  Pa effective stress and -2°C) uniformly through a 2,500 m ice column"

This is a questionable assumption. It would be more accurate to use an effective viscosity appropriate for the mid-column, or at least the mid-lower column (for instance, one quarter or one third of the ice thickness).

This follows the same assumption as in Fowler (2013), so we have updated the text here to say "following Fowler (2013)", but this is largely an illustrative calculation to set the stage. Separately, it further demonstrates the shortcomings of trying to answer this question only using analytical methods.

**2.28.** Appendix A2:**

"an initial approximation of ice rigidity is made based on ice temperature." Ice temperature from where?

Updated to "based on initialised ice temperature (details in Larour2012ContinentalISSM)"

**2.29.** Figure S2:**

This figure needs some work to make it a better visualization. It is way too dark and I really get no information from it at all.

We have added an additional zero-transparency panel to Fig. A2 to make this much clearer, thank you for pointing this out.

**2.30.** Figure S3:**

As I mentioned in my major comments, using a single vertically constant value for effective stress is going to underestimate the vertical gradient in viscosity. Also, the caption should use "vertical exaggeration", not "vertical enhancement".

We have switched this from enhancement to exaggeration. Please see our repsonse to Point 2.1 regarding the constant effective stress value.

**2.31.** Figure S5:**

As I mentioned elsewhere, I want to see an additional figure showing the strain rate components (and overall magnitude) in addition to the velocity components.

Please see our response to Point 2.12.

**2.32.** Figure S7:**

This shows the average basal heat flux across the whole model domain, right? Your model has substantial local variation in the temperature structure, which should also produce local variability in the basal heat flux. It would be good if the caption explicitly stated that this is a spatial average of heat flux.

**Updated to read "Domain-averaged heat flux"**

It is also worth noting two things here: 1) the initial steep decrease in basal heat flow is likely because your initial temperature field is not in equilibrium with your imposed boundary conditions. If you started with a steady 1D thermal profile (or at least, a reasonable approximation of a steady 1D profile), then this initial adjustment would be much smaller.

2) You have imposed a constant basal temperature because that is standard in convection modeling. However, an ice sheet bed can only be held at a constant temperature if there is water present. If the basal heat flux increases because of convection, and there is no water present, then the basal temperature will drop, thus weakening the convection. Alternately, if there is water present, then the increase in heat flow will lead to freeze-on. If sufficient water supply is available (an important limitation), then the extra heat flow from convection will be provided by the latent heat of freezing water.

These are valid points. Please see our response to Points 2.3 and 2.4.